# GARLIC: Graph Attention-based Relational Learning of Multivariate Time Series in Intensive Care

**Yanke Li**[*]
SCAI Lab, D-HEST
ETH Zürich & SPF
yanke.li@hest.ethz.ch

**Ruirui Wang**[*]
Department of Informatics
University of Zürich
wangruirui45@outlook.com

**Manuel Günther**
Department of Informatics
University of Zürich
guenther@ifi.uzh.ch

**Diego Paez-Granados**
SCAI Lab, D-HEST
ETH Zürich & SPF
diego.paez@hest.ethz.ch

## Abstract

Healthcare data, such as Intensive Care Unit (ICU) records, comprise heterogeneous multivariate time series sampled at irregular intervals with pervasive missingness. However, clinical applications demand predictive models that are both accurate and interpretable. We present our Graph Attention-based Relational Learning for Intensive Care (GARLIC) model, a novel neural network architecture that imputes missing data through a learnable exponential-decay encoder, captures inter-sensor dependencies via time-lagged summary graphs, and fuses global patterns with cross-dimensional sequential attention. All attention weights and graph edges are learned end-to-end to serve as built-in observation-, signal-, and edge-level explanations. To reconcile auxiliary reconstruction and primary classification objectives, we developed an alternating decoupled optimization scheme that stabilizes training. On three ICU benchmarks (PhysioNet 2012 & 2019, MIMIC-III), GARLIC sets the new state of the art in outcome prediction, significantly improving AUROC and AUPRC over best-performing baselines at comparable computational cost. Ablation studies confirm the contribution of each module, and feature-removal trials validate the fidelity of importance attribution through a monotonic performance drop (full > top 50% > random 50% > bottom 50%). Real-time case studies demonstrate actionable risk warnings with transparent explanations, marking a significant advance toward accurate, explainable deep learning for irregularly sampled ICU time series data. Moreover, we demonstrated GARLIC's superiority in data imputation and classification on various time-series datasets beyond the ICU domain, showing its generalizability and applicability to broader tasks.

## 1 Introduction

Continuous monitoring of critically ill patients in Intensive Care Units (ICU) generates vast streams of multivariate time series data, vital signs, laboratory results, and therapeutic interventions, which are irregularly sampled and rife with missing values. Effective analysis of this data promises timely detection of deterioration and personalized treatment, yet two challenges impede progress. First, the inherent irregularity and heterogeneity of clinical measurements render standard sequence models (*e. g.*, RNNs, Transformers) suboptimal: naïve imputation biases downstream tasks, while specialized approaches (*e. g.*, GRU-D (Che et al., 2018), Latent-ODEs (Chen et al., 2018), mTAND (Shukla & Marlin, 2021)) often ignore inter-sensor dependencies. Second, interpretability is paramount in healthcare, but prevalent post-hoc methods (*e. g.*, Integrated Gradients (Duell et al., 2023), SHAP (Lundberg & Lee, 2017)) demand extra computation and can yield inconsistent explanations, and even inherently interpretable architectures (*e. g.*, RETAIN (Choi et al., 2016), shapelet models (Li et al., 2021; Wen et al., 2025)) either sacrifice performance or struggle with irregular sampling.

---

[*]Equal contribution

In this work, we introduce Graph Attention-based Relational Learning for Intensive Care (GARLIC), a unified framework displayed in figure 1 that simultaneously addresses irregular sampling, missingness, and the need for transparent decision making. GARLIC models each signals dynamics via a decay-based latent feature encoder, constructs time-lagged summary graphs to capture inter-signal relationships on signal horizons, and refines representations through a cross-dimensional sequential attention module. Critically, all attention weights and graph structures are learned end-to-end, yielding built-in explanations at the level of individual time steps, particular sensors, and their interactions, without degrading classification accuracy or efficiency in predicting clinical outcomes.

Our contributions are threefold: **Methodologically,** we propose a multistage graph-attention architecture that integrates (i) decay-based imputation for irregular missingness, (ii) self-attentive, time-lagged graph learning to exploit spatial dependencies, and (iii) cross-dimensional attention to fuse signal embeddings over time. Our method also achieves real-time **integrated interpretability.** By compressing learned attention matrices and sparsifying summary graphs via $\mathcal{L}_1$ regularization, GARLIC provides concise, quantitative importance scores for each observation, signal, and edge in the sensor graph, enabling correct contribution estimations for practitioner-friendly explanations without an external explainer. **Empirically,** GARLIC sets a new state of the art, while maintaining computational cost on par with leading baselines on benchmark ICU datasets. Furthermore, we demonstrate GARLIC's superiority in data imputation and classification on various datasets beyond the ICU domain, showing its generalizability and applicability to broader tasks and datasets.

## 2 RELATED WORK

**Irregular Multivariate Time Series Classification.** To handle irregularly sampled multivariate time series, recent methods can be broadly categorized based on how they handle missingness and model temporal structure. Some approaches replace standard imputation with learnable interpolation mechanisms, such as exponential decay or decomposition-based interpolation, allowing more expressive modeling of temporal trends (Che et al., 2018; Chen et al., 2018; Shukla & Marlin, 2019). Others forgo explicit imputation by modeling continuous-time latent dynamics using neural ODEs or stochastic processes, sometimes integrating them with attention mechanisms (Rubanova et al., 2019; Chen et al., 2018; Jin et al., 2023; Chen et al., 2023; Park et al., 2025). Attention-based architectures further exploit temporal information via time-aware encoding, adaptive aggregation over observed values, or learnable gating schemes tailored to irregular inputs (Horn et al., 2020; Tipirneni & Reddy, 2022; Shukla & Marlin, 2021; Huang et al., 2024; Wang et al., 2023; Zheng et al., 2024; Zhang et al., 2023). Some methods transform time series into alternative structures, such as images or graphs, to leverage architectures such as vision transformers or message-passing networks (Li et al., 2023; Zhang et al., 2022). Inter-signal dependencies, though commonly used during classification, are rarely exploited during missingness modeling. For example, RAINDROP (Zhang et al., 2022) claims to learn such dependencies, yet both our results and the prior analysis (Zheng et al., 2024) show that its best performance comes from fixed graphs, indicating a limited benefit from graph learning. In contrast, GARLIC explicitly leverages both temporal and inter-signal dependencies during missingness modeling, leading to more effective handling of irregular time series.

**Explainable Models for Time Series.** Explainable methods in time series analysis are broadly categorized into post-hoc explainers and self-interpretable models (Zhao et al., 2023). Post-hoc approaches such as gradient-based attribution methods (Guo et al., 2019; Duell et al., 2023), perturbation-based techniques (Kashiparekh et al., 2019), and surrogate modeling via LIME or SHAP (Ribeiro et al., 2016; Lundberg & Lee, 2017) aim to generate explanations after model training, often through backpropagation or input masking. Although these methods are model-agnostic and widely applicable, they require additional computation and may yield inconsistent attributions across similar inputs. Self-interpretable models aim to embed transparency directly into the training and inference process. Shapelet-based approaches (Li et al., 2021; Ma et al., 2020b; Younis et al., 2024; Qu et al., 2024; Wen et al., 2025) learn prototypical subsequences that act as human-readable components for classification. Despite offering intuitive visualization, these methods are primarily validated on regularly sampled physiological data such as EEG or ECG (Wen et al., 2025; Abdullah et al., 2023; Neves et al., 2021), and their interpretability remains hard to quantify objectively. Attention-based models provide more flexible explanations by interpreting learned attention weights or recalibration mechanisms (Choi et al., 2016; Guo et al., 2019; Qin et al., 2017; Ma et al., 2020a). However, most existing methods assume regular sampling and fully observed inputs, limiting their reliability on irregularly sampled

medical data, where unmodeled temporal gaps can introduce explanation bias. These limitations motivate the need for more interpretable methods that remain robust in real-world conditions. GARLIC addresses this need by incorporating interpretable modeling into the missingness-handling process, enabling faithful explanations in the presence of irregularity and missingness.

# 3 PROPOSED METHOD

## 3.1 PROBLEM DEFINITION

We consider the task of predicting clinical outcomes from multivariate time series sampled irregularly in intensive care settings. Formally, let $\mathcal{D} = \left\{ (S_n, y_n) \right\}_{n=1}^{N}$ be a labeled dataset of $N$ patient episodes, where $y_n \in \{1, \ldots, C\}$ is the outcome label for episode $S_n$. Each episode $S_n$ consists of asynchronous observations of $K$ clinical signals (e.g., vital signs, laboratory tests), aligned to a timeline of up to $T$ discrete time steps. We represent each episode as an input matrix $\mathbf{X} \in \mathbb{R}^{K \times T}$, where $x_{k,t}$ stores the value of signal $k$ at time $t$, and a corresponding binary mask $\mathbf{M} \in \{0, 1\}^{K \times T}$ indicating whether the value is observed. These signals exhibit heterogeneous statistical properties and are sampled at irregular intervals, leading to misaligned time axes and pervasive missingness. Our objective is to learn a model that (i) accurately predicts each $y_n$ from the irregular multivariate time series $S_n$ and (ii) provides transparent attributions at the observation-level to foster clinical trust.

## 3.2 GARLIC

Figure 1 shows our proposed modular framework GARLIC for the prediction of clinical outcomes that combines local reconstruction with global reasoning. Specifically, graph-based message passing reconstructs missing features from local temporal and inter-signal context, while cross-dimensional sequential attention captures global dependencies across time and signals for accurate prediction. To align the auxiliary reconstruction task with the primary prediction objective and stabilize the training, we employ an alternating decoupled optimization strategy that explicitly decouples these two tasks.

### 3.2.1 LATENT FEATURE MODELING

Clinical time series are sparse, irregular, and heterogeneous, so naïve imputation (*e. g.*, mean or zero fill) can distort signal-specific dynamics. We therefore introduce a *latent feature modeling* stage that combines time-aware imputation with per-signal encoding. We adopt an exponential-decay mechanism (Che et al., 2018) to model the diminishing relevance of past observations. Given the elapsed time $\Delta_t$ since the last observation, the decay factor $\gamma_t$ and the imputed value $\hat{x}_{k,t}$ at time $t$ are defined as:

$$\hat{x}_{k,t} = \gamma_t x_{k,t'} + (1 - \gamma_t)\bar{x}_k \quad \text{with} \quad \gamma_t = \exp\left\{-\max(0, w_k \Delta_t + b_k)\right\} \tag{1}$$

where $w_k$ and $b_k$ are learnable parameters, $x_{k,t'}$ is the most recent observed value, and $\bar{x}_k$ is the empirical mean. We then construct an augmented input at each time step to retain both the observed or imputed value and the missingness indicator $m_{k,t} \in \{0, 1\}$:

$$\tilde{\mathbf{x}}_{k,t} = [x_{k,t} \cdot m_{k,t} + \hat{x}_{k,t} \cdot (1 - m_{k,t}), \; m_{k,t}]^\top. \tag{2}$$

Given the heterogeneity of clinical signals, we apply a dedicated two-layer MLP to each augmented input $\tilde{\mathbf{x}}_{k,t}$, yielding latent embeddings $\mathbf{z}_{k,t} \in \mathbb{R}^D$:

$$\mathbf{z}_{k,t} = \text{MLP}_k\left(\tilde{\mathbf{x}}_{k,t}\right). \tag{3}$$

This signal-specific encoding enables the model to capture the distinct statistical scale, dynamics, and semantics of each clinical signal, which would be obscured under a shared encoder.

### 3.2.2 TIME-LAGGED GRAPH MESSAGE PASSING

To reconstruct missing values in clinical time series, we introduce a local graph-based module that exploits the inherent locality of physiological signals, where each signal's value is strongly influenced by its recent temporal history and interactions with related signals. This temporal and inter-signal locality motivates a design that restricts inference to a short-range window, in contrast

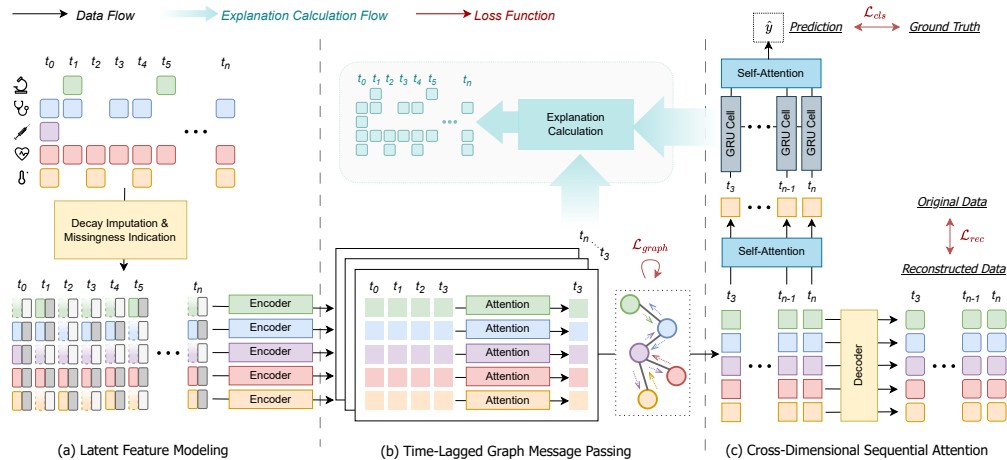

Figure 1: Overview of GARLIC. In the first stage, irregularly-sampled inputs are imputed into a signal-specific latent space. Then, inter-signal dependencies are modeled through a learnable graph-based message passing system, enabling the reconstruction of the original observations. Attention through time and signals enables a GRU model to learn dependencies on a medium time scale, to provide interpretable predictions.

to the global modeling used later for classification. Specifically, given latent embeddings $\mathbf{z}_{k,t}$, we extract a temporal window of length $\tau + 1$ for each signal, where $\tau$ controls the size of the local context, add sinusoidal positional embeddings $\text{PE}(j)$, and apply signal-wise lag attention:

$$\bar{\mathbf{e}}_{k,t} = \text{LagAttn}_k\left(\{\mathbf{z}_{k,j} + \text{PE}(j)\}_{j=t-\tau}^t\right) = \sum_{j=t-\tau}^t \beta_{k,j,t}\mathbf{v}_{k,j},$$

$$\beta_{k,j,t} = \frac{\exp\left(\mathbf{q}_{k,t}^\top\mathbf{k}_{k,j}/\sqrt{D}\right)}{\sum_{j'=t-\tau}^t \exp\left(\mathbf{q}_{k,t}^\top\mathbf{k}_{k,j'}/\sqrt{D}\right)}, \tag{4}$$

with $\mathbf{v}_{k,j} = \mathbf{W}_v(\mathbf{z}_{k,j} + \text{PE}(j))$, $\mathbf{k}_{k,j} = \mathbf{W}_k(\mathbf{z}_{k,j} + \text{PE}(j))$, $\mathbf{q}_{k,t} = \mathbf{W}_q(\mathbf{z}_{k,t} + \text{PE}(t))$ , and $\bar{\mathbf{e}}_{k,t} = \sum_{j=t-\tau}^t \beta_{k,j,t}\mathbf{v}_{k,j}$ is the window-level representation for signal $k$ at time $t$, with $\beta_{k,j,t}$ being the corresponding attention weight. We stack the outputs for all $K$ signals to form $\bar{\mathbf{E}}_t \in \mathbb{R}^{K \times D}$, which summarizes contextual information on the temporal window for each signal.

To model interactions among signals at a given temporal lag $\tau$, we construct a summary graph $\mathcal{G}_\tau$ defined by a learnable adjacency matrix $\mathbf{W}_\tau \in \mathbb{R}^{K \times K}$, where each node represents a signal and edges encode pairwise dependencies. Graph-based message passing is then applied to propagate information across correlated signals:

$$\mathbf{H}_t = \mathbf{W}_\tau \bar{\mathbf{E}}_t \in \mathbb{R}^{K \times D}, \tag{5}$$

where $\mathbf{H}_t$ denotes the time-$t$ representations of all signals, updated based on historical context and relational dependencies from the graph $\mathcal{G}_\tau$. By decoupling attention-based temporal encoding from graph propagation, this module flexibly captures short-range dynamics and inter-signal dependencies. Moreover, varying the lag $\tau$ enables adaptive multi-scale modeling of physiological dependencies. We treat $\tau$ as a hyperparameter, with ablation results in Appendix D.2.

### 3.2.3 CROSS-DIMENSIONAL SEQUENTIAL ATTENTION

To construct a global representation for classification, we apply a two-stage attention cascade to the signal-wise encoding $\{\mathbf{H}_t\}_{t=\tau}^T$ produced by the message passing module. At each time step $t$, we first apply an attention mechanism $\text{SignalAttn}$ over the $K$ signals to obtain a weighted aggregation:

$$\bar{\mathbf{u}}_t = \text{SignalAttn}(\mathbf{H}_t) = \sum_{k=1}^K \alpha_{k,t}^{\text{sig}}\mathbf{u}_{k,t} \qquad \alpha_{k,t}^{\text{sig}} = \frac{\exp\left((\mathbf{q}_t')^\top\mathbf{k}_{k,t}'/\sqrt{D}\right)}{\sum_{j=1}^K \exp\left((\mathbf{q}_t')^\top\mathbf{k}_{j,t}'/\sqrt{D}\right)} \tag{6}$$

with $\mathbf{u}_{k,t} = \mathbf{W}'_v \mathbf{h}_{k,t}$, $\mathbf{k}'_{k,t} = \mathbf{W}'_k \mathbf{h}_{k,t}$, $\mathbf{q}'_t = \mathbf{W}'_q \bar{\mathbf{h}}_t$, while $\alpha^{\mathrm{sig}}_{k,t}$ are the learned attention weights and $\bar{\mathbf{u}}_t$ summarizes the cross-signal information at time $t$. To capture temporal dependencies, the sequence $\{\bar{\mathbf{u}}_t\}_{t=\tau}^{T}$ is processed by a Gated Recurrent Unit (GRU):

$$\mathbf{g}_t = \mathrm{GRU}(\bar{\mathbf{u}}_t, \mathbf{g}_{t-1}), \tag{7}$$

which models local temporal continuity, reflecting gradual transitions in clinical states. To complement the GRU's locality bias and enable direct modeling of long-range dependencies, we apply temporal self-attention over the GRU outputs, incorporating sinusoidal positional encoding $\mathrm{PE}(t)$:

$$\mathbf{Y} = \mathrm{TemporalAttn}\big(\{\mathbf{g}_{t'} + \mathrm{PE}(t')\}_{t'=\tau}^{T}\big) \in \mathbb{R}^{(T-\tau+1)\times D} = \left\{ \sum_{t'=\tau}^{T} \alpha^{\mathrm{time}}_{t',t} \mathbf{y}_{t'} \right\}_{t=\tau}^{T}$$

$$\alpha^{\mathrm{time}}_{t',t} = \frac{\exp\left((\mathbf{q}''_t)^\top \mathbf{k}''_{t'}/\sqrt{D}\right)}{\sum_{s=\tau}^{T} \exp\left((\mathbf{q}''_t)^\top \mathbf{k}''_s/\sqrt{D}\right)} \tag{8}$$

with $\mathbf{y}_{t'} = \mathbf{W}''_v(\mathbf{g}_{t'} + \mathrm{PE}(t'))$, $\mathbf{k}''_{t'} = \mathbf{W}''_k(\mathbf{g}_{t'} + \mathrm{PE}(t'))$, $\mathbf{q}''_t = \mathbf{W}''_q(\mathbf{g}_t + \mathrm{PE}(t))$ being the value, key, and query embeddings at time $t'$, and $\alpha^{\mathrm{time}}_{t',t}$ is the corresponding attention weight. Finally, we average-pool $\mathbf{Y}$ over time and pass the result through a classification head and softmax to obtain class predictions $\hat{y} \in \mathbb{R}^C$ for all $C$ classes. This attention-based pipeline captures both signal-level importance and temporal saliency, supporting accurate downstream classification.

**Comparison to Prior Work.** Our design is inspired by interpretable time-series models such as RETAIN (Choi et al., 2016) and IMV-LSTM (Guo et al., 2019), which combine recurrence with attention to capture temporal dynamics and signal relevance. RETAIN applies attention at both the timestep and signal levels but aggregates signal contributions uniformly across time, limiting its ability to localize transient patterns. IMV-LSTM applies signal-level attention to the hidden states produced by per-signal LSTMs, which aggregate temporal information before computing importance, potentially obscuring instantaneous signal saliency. In contrast, GARLIC applies signal-level attention directly at each time step *before* any recurrent modeling, ensuring that the importance of fine-grained signals is preserved and explicitly captured prior to temporal fusion.

### 3.3 INTERPRETABILITY AND ATTRIBUTION COMPUTATION

Interpretability quantifies how transparently a models decisions can be understood by humans (Zhang et al., 2021). GARLIC provides local interpretability by attributing each prediction directly to its input features. We compute input-level attribution scores by tracing learned importance weights backward from the model output. This backward trace mirrors the forward path through the model's three core components as illustrated in figure 1 and described in section 3.2.

**Attention-based saliency (section 3.2.3);** Let $\{\alpha^{\mathrm{sig}}_{k,t}\}_{k=1,t=\tau}^{K,T}$ denote the signal-level attention scores of equation 6, and $\{\alpha^{\mathrm{time}}_{t,t'}\}_{t,t'=\tau}^{T}$ the temporal attention scores of equation 8. We define the joint saliency as the element-wise product:

$$s_{k,t'} = \sum_{t=\tau}^{T} \alpha^{\mathrm{time}}_{t',t} \cdot \alpha^{\mathrm{sig}}_{k,t} \tag{9}$$

which yields a relevance matrix $\mathbf{S} \in \mathbb{R}^{K \times (T-\tau+1)}$ capturing the importance of each signal-time pair.

**Graph-based propagation (section 3.2.2).** We propagate the relevance scores $\mathbf{S}$ across signals using the transposed time-lagged graph $\mathbf{W}_\tau \in \mathbb{R}^{K \times K}$ from equation 5, and redistribute them temporally using the attention weights $\{\beta_{k,t,j}\}_{k=1,t=\tau,j=0}^{K,T,\tau}$ from equation 4. The resulting importance scores $a_{k,t}$ denote the propagated relevance of signal $k$ at time $t$ and are calculated as:

$$a_{k,t} = \sum_{j=0}^{\tau} \left[ (\mathbf{W}_\tau^\top \mathbf{S})_{k,t+j} \cdot \beta_{k,t+j,\tau-j} \right]. \tag{10}$$

**Observation masking and redistribution (section 3.2.1).** To isolate contributions of each observation, we apply the binary mask $\mathbf{M} \in \{0,1\}^{K \times T}$. Since decay-based imputation blends multiple past inputs, attribution to unobserved entries is ill-defined. We approximate it by redistributing their contribution uniformly across observed positions of the same signal:

$$a^{\mathrm{final}}_{k,t} = a_{k,t} \cdot m_{k,t} + (1 - m_{k,t}) \frac{\sum_{t'=1}^{T} a_{k,t'} \cdot (1 - m_{k,t'})}{\sum_{t'=1}^{T} m_{k,t'} + \epsilon}, \tag{11}$$

where $\epsilon > 0$ ensures numerical stability. The final attribution map $\mathbf{A}^{\text{final}} \in \mathbb{R}^{K \times T}$ assigns salience to individual observations in a way that is fully consistent with the models forward computation.

## 3.4 TRAINING STRATEGY

### 3.4.1 LOSS FUNCTION

During training, we make use of both an auto-encoder to learn representative features and to impute reasonable values, and the classification head to predict clinical outcomes such as mortality or sepsis onset. Our training objective jointly enforces (i) accurate data reconstruction, (ii) a sparse, interpretable graph, and (iii) strong downstream classification performance, via the following components:

**(i) Reconstruction Loss ($\mathcal{L}_{\text{rec}}$).** After message passing, the fused hidden states $\mathbf{H}$ are passed through a decoder to reconstruct the input data. Let $\hat{x}_{k,t}$ denote the reconstructed values, $x_{k,t}$ the observed data, and $m_{k,t} \in \{0, 1\}$ the binary mask indicating observed entries. The reconstruction objective minimizes the masked mean squared error.

**(ii) Graph Regularization Loss ($\mathcal{L}_{\text{graph}}$).** To capture inter-signal dependencies, the model learns a lag-$\tau$ dependency graph $\mathcal{G}_\tau$ with adjacency matrix $\mathbf{W}_\tau \in \mathbb{R}^{K \times K}$, where each entry denotes the influence between signals. To ensure that the learned graph highlights only the most salient dependencies, we apply an $\ell_1$ regularization to promote sparsity. This encourages the model to focus on a small set of essential edges, thereby improving interpretability and mitigating overfitting from spurious or redundant connections.

**(iii) Classification Loss ($\mathcal{L}_{\text{cls}}$).** For prediction, we use categorical cross-entropy between the model's output $\hat{y} \in \mathbb{R}^C$ and the ground-truth label $y \in \{1, \dots, C\}$.

**Total Loss.** We combine the above terms into a single objective:

$$\mathcal{L} = \underbrace{\sum\nolimits_{k,t} m_{k,t}(x_{k,t} - \hat{x}_{k,t})^2}_{\mathcal{L}_{\text{rec}}} + \lambda_g \underbrace{\|\mathbf{W}_\tau\|_1}_{\mathcal{L}_{\text{graph}}} - \lambda_c \underbrace{\log \hat{y}_y}_{\mathcal{L}_{\text{cls}}}, \tag{12}$$

where $\lambda_g$ and $\lambda_c$ balance the regularization and supervision terms. This multi-objective formulation enables the model to exploit structural priors and label supervision jointly, improving both reconstruction fidelity and predictive performance.

### 3.4.2 ALTERNATING DECOUPLED OPTIMIZATION

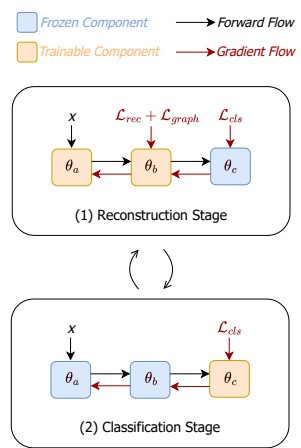

Figure 2: Alternating decoupled optimization training.

Joint optimization of reconstruction and classification often leads to gradient interference due to their differing objectives: reconstruction seeks faithful input recovery, while classification prioritizes discriminative features. To mitigate this conflict, we adopt an *alternating decoupled optimization* strategy, inspired by De-FRCN (Qiao et al., 2021), to isolate learning dynamics between modules. As illustrated in figure 2, our model consists of three parametrized components $\theta_a$, $\theta_b$, and $\theta_c$ corresponding to: (a) Latent Feature Modeling (section 3.2.1), (b) Time-Lagged Graph Message Passing (section 3.2.2), and (c) a Cross-Dimensional Sequential Attention classifier (section 3.2.3). In **Stage 1**, we update shared modules ($\theta_{a,b}$) via a composite loss of reconstruction, graph regularization, and classification, while fixing the classifier ($\theta_c$):

$$\theta_{a,b} \leftarrow \theta_{a,b} - \eta \nabla_{\theta_{a,b}} (\mathcal{L}_{\text{rec}} + \lambda_g \mathcal{L}_{\text{graph}} + \lambda_c \mathcal{L}_{\text{cls}}). \tag{13}$$

In **Stage 2**, we freeze $\theta_{a,b}$ and update $\theta_c$ using the classification loss:

$$\theta_c \leftarrow \theta_c - \eta \nabla_{\theta_c} \mathcal{L}_{\text{cls}}. \tag{14}$$

This decoupling reduces representational interference, stabilizes training, and improves classification performance. Full details and reasoning are provided in Appendix A.3.

# 4 EXPERIMENTS

## 4.1 EXPERIMENTAL SETUP

**Datasets.** We evaluate our method on three widely used clinical time series benchmarks: MIMIC-III (Johnson et al., 2016), PhysioNet Challenge 2012 (P12) (Silva et al., 2012), and PhysioNet Challenge 2019 (P19) (Reyna et al., 2019). These datasets feature diverse patient populations, varying recording frequencies, and significant missingness, reflecting the challenges of real-world ICU records. All tasks are framed as binary classifications: in-hospital mortality for MIMIC-III and P12, and sepsis onset for P19. Detailed preprocessing procedures and statistics are provided in Appendix B.

**Baselines.** We compare our method against strong baselines across two categories. Irregularity-aware models encompass: variants of RNNs (RNN-Mean, RNN-Decay, and RNN-$\Delta_t$; details in Appendix C.1), GRU-D (Che et al., 2018), ODE-RNN (Rubanova et al., 2019), L-ODE-RNN (Chen et al., 2018), L-ODE-ODE (Chen et al., 2018), mTAND (Shukla & Marlin, 2021), Warpformer (Zhang et al., 2023), and MTSFormer (Zheng et al., 2024), which explicitly model missing values, irregular sampling, or continuous-time dynamics. This category also includes MTGNN (Wu et al., 2020) and Raindrop (Zhang et al., 2022), which further incorporate inter-signal dependencies through learned or predefined graphs. Interpretable models are: RETAIN (Choi et al., 2016), IMV-LSTM (Guo et al., 2019), and DARNN (Qin et al., 2017), which aim to provide signal- and temporal-level interpretability. Implementation details and hyperparameter settings for all baselines are given in Appendix C.1.

**Implementation Details.** Each dataset is split into training, validation, and test sets in an 8:1:1 ratio, with stratified sampling on both outcome labels and key patient characteristics (*e. g.*, gender, ICU type) to preserve class balance. Hyperparameters are chosen via grid search on the validation split, and for baselines, we mirror the tuning ranges and protocols of their sources to ensure a fair comparison; detailed settings are provided in Appendix C.1. Models are trained for up to 100 epochs with Adam optimizer (learning rate tuned per dataset), employing early stopping with a patience of 10 epochs based on the validation loss. We report both AUROC and AUPRC to account for class imbalance, and all results are averaged over five trials with distinct random seeds. All experiments were conducted on a server with an Intel Xeon Silver 4314 processor (16 cores, 2.40GHz), 256GB RAM, and dual NVIDIA GeForce RTX 3090 GPUs (24GB VRAM each). All training experiments utilize a single GPU configuration.

Table 1: Model performance in terms of AUROC and AUPRC. For both metrics, we report mean $\pm$ std in % over five random seeds. The first block lists strong baselines designed for irregular time series, the second block includes interpretable models, and the final block shows results for our proposed method. We highlight the **best** and the second-best results per column.

| Models | P12 | | P19 | | MIMIC-III | |
|---|---|---|---|---|---|---|
| | **AUROC** | **AUPRC** | **AUROC** | **AUPRC** | **AUROC** | **AUPRC** |
| RNN-$\Delta_t$ | 82.46±1.57 | 47.29±4.33 | 89.21±0.63 | 51.20±2.50 | 82.98±2.27 | 50.31±4.22 |
| RNN-Mean | 82.38±2.11 | 47.80±3.98 | 88.08±0.88 | 45.80±2.10 | 84.18±1.70 | 50.06±5.17 |
| RNN-Decay | 84.26±1.10 | 49.81±3.31 | 89.15±1.15 | 49.73±3.46 | 87.63±0.79 | 56.29±3.29 |
| GRU-D | 84.45±1.90 | 50.74±4.96 | 88.73±0.86 | 47.81±3.37 | 86.36±1.68 | 56.14±1.87 |
| ODE-RNN | 83.02±1.72 | 48.64±2.68 | 89.97±0.86 | 54.29±2.76 | 88.07±0.77 | 61.03±2.00 |
| L-ODE-RNN | 80.59±2.06 | 41.40±4.15 | 89.63±1.31 | 51.02±3.77 | 86.98±0.58 | 56.88±2.20 |
| L-ODE-ODE | 83.93±1.14 | 49.55±1.43 | 90.01±2.04 | 53.54±6.12 | 87.36±0.94 | 59.09±1.93 |
| MTGNN | 80.95±1.71 | 43.61±4.93 | 86.94±1.65 | 40.97±5.26 | 86.24±0.67 | 51.74±0.92 |
| mTAND | 84.30±1.69 | 50.05±3.96 | 81.73±1.53 | 37.27±3.93 | 88.00±0.31 | 57.73±2.21 |
| RAINDROP | 83.03±1.37 | 45.91±3.55 | 87.41±1.13 | 46.33±3.09 | 87.18±1.51 | 57.06±6.01 |
| Warpformer | 84.88±0.48 | 50.62±1.25 | 89.95±0.57 | 54.10±2.38 | 89.17±0.57 | 61.52±1.93 |
| MTSFormer | 83.65±1.70 | 50.31±4.30 | 87.88±0.61 | 48.80±2.97 | 88.14±0.50 | 61.09±0.95 |
| IMV-LSTM | 84.02±1.30 | 49.08±3.28 | 84.80±2.76 | 42.87±8.27 | 86.85±1.33 | 54.96±3.61 |
| DARNN | 79.84±1.36 | 42.70±2.09 | 74.36±3.50 | 20.89±2.73 | 82.56±1.25 | 46.33±4.08 |
| RETAIN | 83.08±1.11 | 49.27±3.01 | 78.09±2.22 | 26.04±1.35 | 82.40±0.94 | 46.55±1.96 |
| GARLIC | **86.40±0.86** | **56.89±1.75** | **90.96±0.84** | **55.29±2.45** | **90.09±0.45** | **64.85±1.68** |

## 4.2 MAIN RESULTS

Table 1 reports the AUROC and AUPRC of all methods on three benchmark datasets. GARLIC consistently achieves the highest scores, setting the new state-of-the-art performance. The improvement is especially notable on the P12 dataset, which contains substantial missingness (statistics in Appendix B), demonstrating the models robustness to sparse and irregular multivariate time series. Compared to other self-interpretable models with attention mechanisms—such as RETAIN, DARNN, and IMV-LSTM—GARLIC achieves substantially higher predictive accuracy while preserving interpretability. Compared to irregularity-aware models that primarily address temporal gaps and graph-based models like MTGNN and Raindrop that rely on static graphs without temporal modeling, GARLIC achieves better performance consistently by dynamically capturing time-lagged dependencies across signals. Apart from ICU outcome prediction, we also prove GARLIC's capability in data imputation and human activity recognition, showcasing its generalizability and applicability to broader tasks and datasets. More details can be found in Appendix E.

**Ablation Study.** To assess the contribution of each component in GARLIC, we perform a series of ablations on P12 and P19 benchmarks, with results summarized in Appendix D.1. These results demonstrate that each module substantially contributes to GARLICs overall superiority.

**Model Efficiency.** To evaluate the efficiency of GARLIC, we measure its training time and memory footprint on the P12, P19 and MIMIC III datasets, comparing against 15 baseline models, each configured with optimal hyperparameters and trained using the same batch size for fairness (more details including these quantitative comparative results and theoretical analysis are provided in Appendix A.2). As an example shown in figure 3, GARLIC achieves the best performance while maintaining a comparable computational cost on P12 classification task.

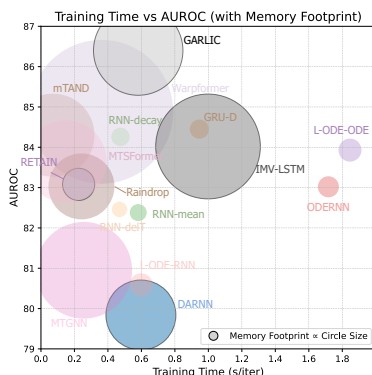

**Graph Learning.** To better understand how our model captures inter-signal dependencies during local reconstruction, we analyze the learned time-lagged summary graph $\mathcal{G}_\tau$, parameterized by the adjacency matrix $\mathbf{W}_\tau$. Preliminary evaluations using an ensemble of advanced AI proxies indicate that the learned graph structures align well with clinical intuition. However, rigorous validation by human medical experts remains an important direction for future work. Further details and discussions are provided in Appendix F.

Figure 3: Comparison of AUROC versus training time for various methods on the P12 dataset.

## 4.3 INTERPRETABILITY EVALUATION

To quantitatively evaluate the reliability of our models attribution scores, we adopt a perturbation-based evaluation strategy inspired by the Remove and Retrain (ROAR) framework (Hooker et al., 2019; Guo et al., 2019; Meng et al., 2022). Specifically, we compare model performance under three masking conditions against the original full-input setting: **(i) Top 50%:** For each sample, we retain the top 50% most important signal-time pairs as determined by attribution scores, masking the remaining inputs. If performance remains close to that of the full-input model, it suggests that the retained inputs capture the primary predictive signal. **(ii) Bottom 50%:** In this condition, we mask the top 50% most important inputs, retaining only the least important half. A marked drop in performance would indicate that the attribution scores correctly identify the most predictive features. **(iii) Random 50%:** As a control, we randomly mask 50% of the inputs, independent of attribution scores. This baseline allows us to disentangle the effect of informed attribution-based masking from random perturbations. In all settings, models are retrained from scratch on the perturbed datasets.

In table 2, we report AUROC and AUPRC (mean $\pm$ std over five seeds) as evaluation metrics. To assess statistical significance, we employ two complementary tests. First, we use the Two One-Sided Tests (TOST) procedure (Schuirmann, 1987) to test whether the performance under the top-50% retention condition is statistically equivalent to the full-input model, with equivalence margins of $\pm 5\%$ for AUROC and $\pm 10\%$ for AUPRC. Secondly, we apply Pages L test (Page, 1963) to examine whether performance follows the expected monotonic ranking: Full $>$ Top-50% $>$ Random-50%

Table 2: Interpretability assessment via input removal. AUROC and AUPRC (%, mean $\pm$ std over five seeds) are reported for models trained on the full input and three 50% subsets of the data (top, random, bottom). Two One-Sided Tests (TOST) p-values evaluate equivalence between full and Top 50% models (margins $\pm$5% AUROC, $\pm$10% AUPRC). Pages L test p-values assess the monotonic ordering (All > Top 50% > Random 50% > Bottom 50%). *p <.05; **p <.005.

| | Models | P12 | | P19 | |
| | | AUROC | AUPRC | AUROC | AUPRC |
|---|---|---|---|---|---|
| **IMV-LSTM** | All | 84.02±1.30 | 49.08±3.28 | 84.80±2.76 | 42.87±8.27 |
| | Top 50% | 76.23±0.32 | 34.46±1.78 | 82.07±1.67 | 36.05±2.91 |
| | Random 50% | 75.23±5.19 | 37.03±8.91 | 76.28±3.73 | 23.31±3.22 |
| | Bottom 50% | 70.12±3.12 | 26.83±2.14 | 79.61±0.88 | 28.59±1.75 |
| | Equivalence (TOST $p$) | 0.9944 | 0.9766 | 0.1024 | 0.0928 |
| | Monotone (Page's L Test $p$) | *0.0011 ** | *0.0084 * | *0.0032 ** | *0.0019 ** |
| **DARNN** | All | 79.84±1.36 | 42.70±2.09 | 74.36±3.50 | 20.89±2.73 |
| | Top 50% | 73.92±10.30 | 35.38±9.83 | 72.17±4.27 | 17.14±3.45 |
| | Random 50% | 65.37±13.23 | 27.25±11.03 | 69.86±4.49 | 17.52±4.21 |
| | Bottom 50% | 67.17±8.38 | 27.05±6.94 | 70.34±6.15 | 15.40±3.38 |
| | Equivalence (TOST $p$) | 0.5661 | 0.3099 | 0.1697 | *0.0004 ** |
| | Monotone (Page's L Test $p$) | *0.0301 * | *0.0132 * | *0.0440 * | *0.0132 * |
| **RETAIN** | All | 83.08±1.11 | 49.27±3.01 | 78.09±2.22 | 26.04±1.35 |
| | Top 50% | 77.04±1.08 | 37.18±3.15 | 74.74±1.64 | 20.17±1.84 |
| | Random 50% | 72.89±5.51 | 34.11±7.87 | 69.08±5.27 | 17.17±3.09 |
| | Bottom 50% | 76.23±1.20 | 36.30±3.68 | 73.28±0.51 | 18.60±2.36 |
| | Equivalence (TOST $p$) | 0.8814 | 0.8172 | 0.1346 | *0.0039 ** |
| | Monotone (Page's L Test $p$) | *0.0132 * | *0.0132 * | *0.0032 ** | *0.0053 * |
| **GARLIC** | All | 86.40±0.86 | 56.89±1.75 | 90.96±0.84 | 55.29±2.45 |
| | Top 50% | 85.60±1.48 | 52.99±3.25 | 87.97±1.24 | 48.24±1.44 |
| | Random 50% | 74.73±6.05 | 35.51±7.04 | 84.73±2.46 | 40.77±6.11 |
| | Bottom 50% | 71.09±1.17 | 33.52±3.42 | 82.52±1.27 | 33.39±2.47 |
| | Equivalence (TOST $p$) | *0.0011 ** | *0.0079 * | *0.0156 * | *0.0399 * |
| | Monotone (Page's L Test $p$) | *0.0002 ** | *0.0004 ** | *0.0002 ** | *0.0002 ** |

> Bottom-50%. Together, these tests quantify both the sufficiency of top-ranked features and the consistency of performance degradation under progressively less informative input subsets. Technical details of both statistical procedures are provided in Appendix G. Across both datasets and metrics, our model passes all eight statistical tests, with all Pages L p-values below 0.005, indicating robust and consistent interpretability. In contrast, three baseline models pass only four or five tests, showing weaker or less consistent attribution quality under identical conditions.

## 4.4 CASE STUDIES

To illustrate GARLIC's interpretability on real ICU trajectories, we analyze two case studies from the P12 and P19 datasets involving standard hemodynamic and metabolic variables (see Appendix H.1). Key indicators include Mean Arterial Pressure (MAP), where values < 65 mmHg imply organ dysfunction, and White Blood Cell (WBC) count, signaling immune activation (Singer et al., 2016).

**Mortality prediction (figure 4a).** For a non-surviving patient, GARLIC appropriately assigns only moderate importance to early, transient heart rate increases. As the episode progresses, the model identifies simultaneous oscillations across multiple blood pressure channels (MAP, systolic, and diastolic) as the primary risk drivers. This specific attribution to synchronous hemodynamic instability aligns with established indicators of clinical deterioration (Silva et al., 2012; Singer et al., 2016), demonstrating the model's sensitivity to complex, multivariate risk factors.

**Sepsis detection (figure 4b).** In a patient developing sepsis, the model successfully filters out early, isolated fluctuations. After hour 12, attribution peaks coincide precisely with a synchronized multi-variable shift: decreasing MAP and bicarbonate (indicating metabolic acidosis), rising HR,

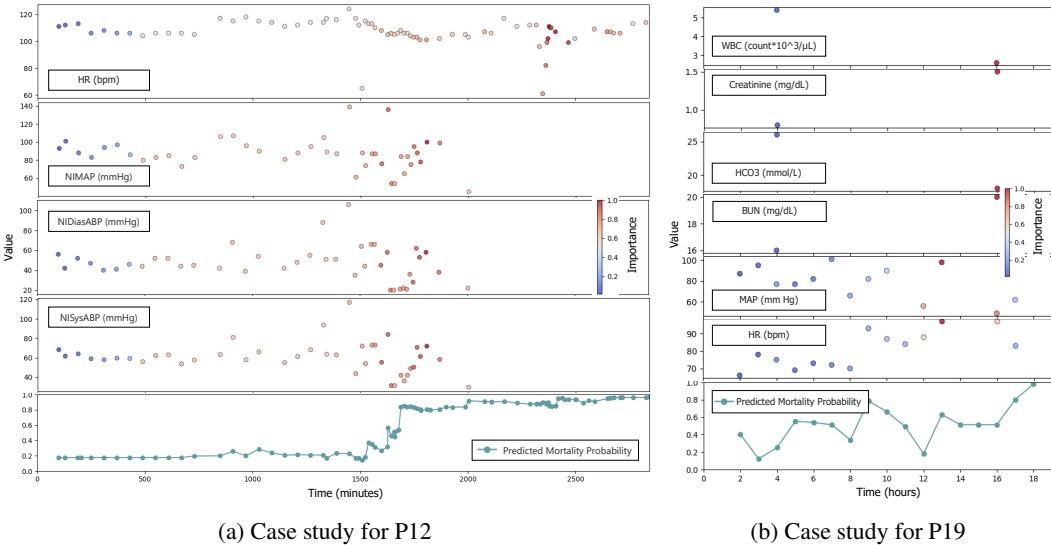

(a) Case study for P12  (b) Case study for P19

Figure 4: Case Study Visualizations. (a) A non-survivor: heart rate (HR), mean arterial pressure (NIMAP), diastolic pressure (NIDiasABP), and systolic pressure (NISysABP) time series with corresponding normalized attributions and the predicted mortality probability over time upon available measurements. (b) A sepsis patient: leukocyte count (WBC), creatinine, bicarbonate ($HCO_3$), blood urea nitrogen (BUN), mean arterial pressure (MAP), and HR time series with their attributions and the predicted sepsis probability over time upon available measurements. Importance scores are normalized within each signal to highlight temporal dynamics.

and elevating kidney function markers (creatinine, BUN). These patterns mirror standard diagnostic criteria for evolving sepsis (Singer et al., 2016; Reyna et al., 2019).

Overall, GARLIC generates plausible attribution maps by highlighting time steps where multiple physiological abnormalities converge, rather than overreacting to isolated noise. Additional case studies are detailed in Appendix H.

## 5 CONCLUSION

We introduced GARLIC, a unified framework for ICU time series that integrates (1) exponential-decay imputation encoding, (2) time-lagged graph message passing, and (3) cross-dimensional sequential attention, trained via alternating decoupled optimization to jointly reconstruct inputs and predict outcomes. On PhysioNet 2012 & 2019 and MIMIC-III, GARLIC achieves the state-of-the-art AUROC and AUPRC. Interpretability evaluation and case study analyses demonstrate that the model produces clinically meaningful and interpretable attributions. Additionally, GARLIC achieved competitive performance in data imputation and classification on various datasets beyond the ICU domain, showing its generalizability and applicability to broader time-series tasks (e.g., human activity classification and imputation). This work advances disease-risk prediction by combining high accuracy with transparent, clinician-friendly explanations, empowering healthcare teams to make more informed decisions and fostering trust in AI-driven insights.

**Limitations and future work.** While GARLIC excels at binary outcome prediction with built-in explanations, several limitations remain that restrict its broader applicability. First, the model has not yet been evaluated for forecasting tasks, which are important in many clinical applications. Next, the reliance on fixed-size sliding windows and discrete time lags may limit its flexibility in modeling irregular or continuous-time signals; we plan to explore adaptive, window-free variants and event-driven formulations. In addition, GARLIC currently excludes static patient features (*e. g.*, demographics, comorbidities) and does not address the severe imbalance in outcomes. We aim to incorporate such features and adopt strategies such as class weighting or resampling. Finally, GARLIC does not yet support streaming or arbitrarily long ICU stays, and its real-world utility hinges on integration with clinician feedback for iterative refinement.

## REPRODUCIBILITY STATEMENT

To facilitate reproducibility, we provide data description and implementation details in section 4.1, including choice of model hyperparameters in Appendix C.2. The implementation codes of GARLIC can be accessed via https://github.com/SCAI-Lab/GARLIC.

## ACKNOWLEDGMENT

This research project was partially supported by the Schweizer Paraplegiker Stiftung and the ETH Zürich Foundation (2021-HS-348) and the JST Moonshot R&D Program, Grant Number JPMJMS2034-18.

## ETHICS STATEMENT

This research did not involve human subjects and, therefore, did not require Institutional Review Board (IRB) approval.

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

# A  ADDITIONAL DETAILS OF THE PROPOSED METHOD

## A.1  ARCHITECTURE DETAILS

### A.1.1  LATENT FEATURE MODELING

**Time-aware imputation.** The imputation module uses an exponential decay mechanism with per-variable parameters $w_k$ and $b_k$, initialized to $0.1$ and $0$, respectively.

**Signal-specific encoders.** Each input variable is processed by a dedicated Multi-Layer Perceptron (MLP), implemented independently for each dimension. Each MLP has two fully connected layers with ReLU activation, both with output dimension $D_f$, which is a shared tunable hyperparameter.

### A.1.2  TIME-LAGGED GRAPH MESSAGE PASSING

**Temporal context encoding.** For each variable, we extract a window of size $\tau+1$ and apply sinusoidal positional encodings before computing attention across the window. This yields a temporally contextualized embedding per variable.

**Graph-based signal interaction.** A time-lagged graph $\mathcal{G}_\tau$ is defined by a learnable adjacency matrix $\mathbf{W}_\tau \in \mathbb{R}^{K \times K}$, initialized as the sum of an identity matrix and a fully connected matrix scaled by a small positive constant. This stabilizes training and introduces a weak prior toward dense connectivity. Message passing is performed via matrix multiplication with $\mathbf{W}_\tau$.

**Lag configuration.** A single lag parameter $\tau$ is shared across variables. Future work may explore variable-specific lags for more flexible modeling.

### A.1.3  CROSS-DIMENSIONAL SEQUENTIAL ATTENTION

**Temporal modeling and fusion.** Signal-level summaries are passed through a GRU with hidden size $D_h$, followed by temporal self-attention over the GRU outputs. Positional encodings are added before attention, and the attended sequence is mean-pooled to obtain a global representation.

**Classifier.** The pooled vector is processed by a two-layer MLP with ReLU activation and hidden dimension $D_h$, producing the final prediction logits.

### A.1.4  TRAINING STRATEGY

**Loss function.** To guide representation learning, the fused representations $\mathbf{H}$ are decoded back to the input space through a two-layer MLP with ReLU activation and hidden size $D_f$. Reconstruction is supervised using masked mean squared error over observed entries. An $\ell_1$ penalty on $\mathbf{W}_\tau$ encourages sparse graph structure, and binary cross-entropy is used for classification. The total loss is a weighted sum of reconstruction loss, graph sparsity regularization, and classification loss, where the latter two are scaled by coefficients $\lambda_g$ and $\lambda_c$, respectively.

**Alternating decoupled optimization.** Parameters are divided into a shared backbone (imputation, encoding, time-lagged attention, message passing) and a prediction head (signal-level attention, GRU, temporal attention, classifier). Each epoch updates only one group while freezing the other. The two groups use separate learning rates and weight decay. Phase transitions are triggered by early stopping with patience, and each phase resumes from the best checkpoint of the previous one.

## A.2  MODEL EFFICIENCY

### A.2.1  THEORETICAL ANALYSIS

We analyze the computational and memory complexity of our model with respect to the sequence length $T$ and the number of variables $K$. The model employs a sliding-window mechanism, where each window spans $\tau+1$ time steps, yielding $L = T - \tau$ valid windows per sequence. Let $D_f$ denote the input feature dimension and $D_h$ the hidden state size. Positional encodings are implemented via table lookups and incur negligible computational overhead.

Table 3: Training time (s/iter) and memory footprint (MB) of baseline models across datasets (batch sizes: P12=64, P19=64, MIMIC-III=16).

| Method | P12 Time | P12 Mem | P19 Time | P19 Mem | MIMIC-III Time | MIMIC-III Mem |
|---|---|---|---|---|---|---|
| GRU-D | 0.946 | 152 | 0.132 | 28 | 1.948 | 187 |
| RNN-mean | 0.581 | 120 | 0.078 | 25 | 1.198 | 88 |
| RNN-decay | 0.475 | 140 | 0.159 | 27 | 0.600 | 87 |
| RNN-delT | 0.469 | 120 | 0.050 | 24 | 0.980 | 76 |
| ODE-RNN | 1.717 | 184 | 0.246 | 32 | 2.737 | 114 |
| L-ODE-RNN | 0.599 | 218 | 0.087 | 37 | 0.650 | 126 |
| L-ODE-ODE | 1.846 | 220 | 0.318 | 38 | 3.873 | 136 |
| mTAND | 0.072 | 2856 | 0.063 | 1007 | 0.205 | 6273 |
| MTSFormer | 0.148 | 2798 | 0.131 | 1653 | 0.107 | 1176 |
| MTGNN | 0.256 | 3864 | 0.185 | 2574 | 0.290 | 2151 |
| Raindrop | 0.242 | 1806 | 0.163 | 313 | 0.128 | 2431 |
| WarpFormer | 0.361 | 8566 | 0.164 | 3981 | OOM | OOM |
| RETAIN | 0.225 | 430 | 0.035 | 59 | 0.353 | 213 |
| DARNN | 0.597 | 2034 | 0.135 | 133 | 0.601 | 612 |
| IMV-LSTM | 0.998 | 4550 | 0.213 | 1977 | 5.284 | 7509 |
| GARLIC | 0.581 | 3322 | 0.184 | 1250 | 0.619 | 1128 |

Table 4: Lightweight GARLIC configurations on P19: varying time lag (feature_dim=32, hidden_size=512)

| Feature Dim | Hidden Size | Time Lag | AUROC (%) | Time (s/iter) | Memory (MB) |
|---|---|---|---|---|---|
| 32 | 512 | 8 | 90.96 | 0.184 | 1250 |
| 32 | 512 | 7 | 90.07 | 0.164 | 1060 |
| 32 | 512 | 6 | 89.51 | 0.153 | 1098 |
| 32 | 512 | 5 | 89.47 | 0.152 | 960 |
| 32 | 512 | 4 | 88.97 | 0.155 | 832 |
| 32 | 512 | 3 | 89.30 | 0.137 | 694 |
| 32 | 512 | 2 | 88.52 | 0.143 | 662 |
| 32 | 512 | 1 | 88.99 | 0.128 | 660 |

The per-layer computational complexity is detailed as follows: the signal-wise encoders require $\mathcal{O}(TKD_f^2)$; the signal-wise time-lagged attention operates at $\mathcal{O}(TKD_f)$ under the assumption that $\tau$ is a small constant; message passing and signal-level attention introduce $\mathcal{O}(TK^2D_f)$ complexity due to all-pairs interactions among variables; the GRU contributes $\mathcal{O}(TD_h^2)$; the temporal attention layer incurs $\mathcal{O}(T^2D_h)$; and the final classification head adds $\mathcal{O}(D_h^2)$.

Aggregating the above components, the overall computational complexity of the model is: $\mathcal{O}(TKD_f^2) + \mathcal{O}(TK^2D_f) + \mathcal{O}(TD_h^2) + \mathcal{O}(T^2D_h) + \mathcal{O}(D_h^2)$. Neglecting the feature and hidden dimensions, the model exhibits an overall complexity of $\mathcal{O}(TK + TK^2 + T + T^2)$.

### A.2.2 EMPIRICAL RESULTS

We benchmark the computational efficiency of GARLIC against fifteen baseline models in terms of training time and memory usage. All models are trained under identical batch sizes with their best hyperparameters to ensure fairness. The results, summarized in table 3, show that GARLIC achieves comparable efficiency while maintaining strong predictive performance.

### A.2.3 PERFORMANCE-EFFICIENCY TRADE-OFF ANALYSIS

We evaluate the impact of model configurations on computational efficiency, focusing on the time lag ($\tau$), feature dimension, and hidden size. The results, presented in Tables 4, 5, 6, 7, 8, and 9, show that reducing these parameters leads to a noticeable decrease in training time and memory usage, although some minor drop in AUROC scores is observed. Smaller configurations, achieved through shorter time lags or reduced feature and hidden dimensions, lower per-iteration runtime and memory

Table 5: Lightweight GARLIC configurations on P19: varying hidden size (feature_dim=32, time_lag=8)

| Feature Dim | Hidden Size | Time Lag | AUROC (%) | Time (s/iter) | Memory (MB) |
|---|---|---|---|---|---|
| 32 | 512 | 8 | 90.96 | 0.184 | 1250 |
| 32 | 256 | 8 | 90.47 | 0.135 | 1186 |
| 32 | 128 | 8 | 90.52 | 0.134 | 1080 |
| 32 | 64 | 8 | 89.62 | 0.133 | 1150 |
| 32 | 32 | 8 | 88.53 | 0.133 | 1140 |

Table 6: Lightweight GARLIC configurations on P19: varying feature dimension (hidden_size=512, time_lag=8)

| Feature Dim | Hidden Size | Time Lag | AUROC (%) | Time (s/iter) | Memory (MB) |
|---|---|---|---|---|---|
| 32 | 512 | 8 | 90.96 | 0.184 | 1250 |
| 16 | 512 | 8 | 90.46 | 0.136 | 710 |
| 8 | 512 | 8 | 89.98 | 0.126 | 566 |

footprint while incurring only a small reduction in predictive performance. These experiments indicate that GARLIC offers flexible trade-offs between efficiency and accuracy, achieving practical computational costs across a wide range of configurations.

## A.3 DECOUPLED OPTIMIZATION RATIONALE

We partition model parameters into three groups for decoupled optimization. The first group, $\theta_a$, includes the time-aware imputation parameters ($w_k$, $b_k$) and the signal-specific MLP encoders, corresponding to the *Latent Feature Modeling* module. The second group, $\theta_b$, comprises the time-lagged attention module with positional encoding and the message passing layer with the learnable adjacency matrix, forming the *Time-Lagged Graph Message Passing* module. The third group, $\theta_c$, consists of the signal-level attention, GRU, temporal self-attention, and the final two-layer MLP classifier, which together compose the *Cross-Dimensional Sequential Attention* module.

Although reconstruction is introduced as an auxiliary objective to improve representation quality by fully leveraging partially observed data, it is not fully aligned with the final classification task. The reconstruction components, *i.e.*, (a) *Latent Feature Modeling* and (b) *Time-Lagged Graph Message Passing* are designed to faithfully recover the input signals, preserving temporal continuity and inter-signal structure. These modules prioritize fidelity to the observed data, independent of its discriminative utility. In contrast, the classification module, *i.e.*, (c) *Cross-Dimensional Sequential Attention* focuses on extracting task-relevant features that effectively differentiate clinical trajectories associated with distinct outcome classes. As a result, it may intentionally discard input patterns that are non-predictive or noisy. This divergence in learning objectives introduces *representational interference* when the system is trained end-to-end, causing unstable updates and suboptimal generalization.

To resolve this issue, we adopt an *alternating decoupled optimization* strategy, an approach generally applicable to multi-objective learning when components serve partially conflicting purposes. Although our overall task is classification, a strong auxiliary reconstruction loss introduces competing gradient signals that, if not properly managed, can impair model performance.

Our strategy alternates between two decoupled stages of optimization. In the first stage, we fix the classifier $\theta_c$ and update the shared modules $\theta_{a,b}$ by minimizing a composite loss that combines reconstruction, graph regularization, and classification objectives:

$$\theta_{a,b} \leftarrow \theta_{a,b} - \eta \nabla_{\theta_{a,b}} \left( \mathcal{L}_{\text{rec}} + \lambda_g \mathcal{L}_{\text{graph}} + \lambda_c \mathcal{L}_{\text{cls}} \right), \quad \text{with } \theta_c \text{ fixed.}$$

In this step, although $\theta_c$ remains static, classification gradients are allowed to flow back through $\theta_{a,b}$ to encourage task-relevant feature extraction during representation learning.

Table 7: Lightweight GARLIC configurations on P12: varying window size (feature_dim=16, hidden_size=128).

| Feature Dim | Hidden Size | Time Lag | AUROC (%) | Time (s/iter) | Memory (MB) |
|---|---|---|---|---|---|
| 16 | 128 | 2 | 86.40 | 0.581 | 3322 |
| 16 | 128 | 1 | 85.18 | 0.221 | 953 |

Table 8: Lightweight GARLIC configurations on P12: varying hidden size (feature_dim=16, time_lag=2).

| Feature Dim | Hidden Size | Time Lag | AUROC (%) | Time (s/iter) | Memory (MB) |
|---|---|---|---|---|---|
| 16 | 128 | 2 | 86.40 | 0.581 | 3322 |
| 16 | 64 | 2 | 86.18 | 0.279 | 1004 |
| 16 | 32 | 2 | 84.37 | 0.282 | 988 |

In the second stage, we freeze $\theta_{a,b}$ and update the classification module $\theta_c$ based solely on classification performance:

$$\theta_c \leftarrow \theta_c - \eta \nabla_{\theta_c} \mathcal{L}_{\text{cls}}, \quad \text{with } \theta_{a,b} \text{ fixed.}$$

By alternating these two steps until convergence, each module is allowed to specialize on its respective objective while still benefiting from shared latent features. This decoupling mitigates conflicting learning dynamics, enhances training stability, and improves overall classification performance.

Our design is inspired by DeFRCN (Qiao et al., 2021), which decouples localization and classification heads in few-shot object detection to avoid multi-task interference. While their setting addresses conflict between distinct tasks, our scenario involves misalignment among sub-objectives within a single classification task due to auxiliary reconstruction. Nevertheless, the underlying principle remains consistent: isolating incompatible optimization signals improves convergence and modular performance.

## B   DATA DETAILS

### B.1   DATASET DESCRIPTION

**MIMIC-III.** We use the Medical Information Mart for Intensive Care III (MIMIC-III) dataset, a large, publicly available database comprising deidentified health-related data from over 40,000 critical care patients admitted to the Beth Israel Deaconess Medical Center between 2001 and 2012 (Johnson et al., 2016). The dataset contains high-resolution clinical information, including demographics, vital signs, laboratory test results, medications, procedures, caregiver notes, imaging reports, and in-hospital as well as post-discharge mortality outcomes. MIMIC-III supports a wide range of research applications in clinical prediction modeling, epidemiology, and decision support, and is notable for its large and diverse ICU population, high temporal granularity, and unrestricted availability to researchers worldwide.

**PhysioNet Challenge 2012 (P12).** The P12 dataset is derived from the PhysioNet/Computing in Cardiology Challenge 2012 (Silva et al., 2012). It comprises clinical records from 12,000 adult ICU stays spanning various unit types, including medical, surgical, trauma, and cardiac ICUs. All patients were admitted for diverse clinical conditions, and ICU stays shorter than 48 hours were excluded. Up to 42 signals are recorded per patient, including 6 admission-level descriptors and 36 time-varying physiological signals (*e. g.*, vital signs and laboratory results). Due to signal sampling frequencies and clinical heterogeneity, the actual number of available signals varies across patients. Each observation is time-stamped with its offset from ICU admission, providing fine-grained temporal resolution for modeling.

**PhysioNet Challenge 2019 (P19).** The P19 dataset is derived from the PhysioNet/Computing in Cardiology Challenge 2019 (Reyna et al., 2019), which focuses on the early prediction of sepsis in ICU settings. It comprises multivariate clinical time series from 40,331 ICU stays, collected across three distinct hospital systems. Each patient record is formatted as an hourly sampled time series with

Table 9: Lightweight GARLIC configurations on P12: varying feature dimension (hidden_size=128, time_lag=2).

| Feature Dim | Hidden Size | Time Lag | AUROC (%) | Time (s/iter) | Memory (MB) |
|---|---|---|---|---|---|
| 16 | 128 | 2 | 86.40 | 0.581 | 3322 |
| 8 | 128 | 2 | 86.26 | 0.198 | 742 |

Table 10: Dataset statistics

| Dataset | Samples | Sensors | Negative:Positive Labels | Missing Ratio (%) | Task |
|---|---|---|---|---|---|
| MIMIC III | 49380 | 103 | 43633:5747 | 98.08±1.10 | Mortality Prediction |
| P12 | 11988 | 36 | 10281:1707 | 94.80±1.51 | Mortality Prediction |
| P19 | 40331 | 34 | 37400:2931 | 85.83±6.13 | Sepsis Prediction |

34 clinical signals, including vital signs, laboratory test results, and demographic attributes. Sepsis annotations follow the Sepsis-3 definition (Singer et al., 2016), requiring both a two-point increase in the Sequential Organ Failure Assessment (SOFA) score and evidence of infection.

### B.2 DATA PREPROCESSING AND STATISTICS

This section details the preprocessing procedures for each dataset and presents the corresponding statistics in table 10 after preprocessing.

**MIMIC-III.** We use the preprocessed version of MIMIC-III (Johnson et al., 2016), following the preprocessing protocol adopted in Warpformer (Zhang et al., 2023). This version contains 103 clinical signals, comprising 61 biomarkers (*e. g.*, vital signs and lab tests) and 42 intervention-related features (*e. g.*, medications and procedures), extracted from 53,423 ICU admissions. Consistent with prior work, we restrict our analysis to adult ICU stays and exclude neonatal cases. For the in-hospital mortality prediction task, we utilize all available time series data collected during the ICU stay, and discard admissions with a length of stay shorter than 24 hours to ensure sufficient temporal context.

**P12.** We follow the official PhysioNet 2012 Challenge structure (Silva et al., 2012) and retain 36 physiological signals after removing static admission descriptors. For each patient, we extract all available timestamped observations from the first 48 hours of ICU stay. Categorical static features (*e. g.*, gender, ICU type) are one-hot encoded. In-hospital mortality outcomes are derived from the provided labels, and corrupted or blacklisted records are excluded during filtering.

**P19.** We follow the official PhysioNet 2019 Challenge setup (Reyna et al., 2019). Each patient record is parsed to extract hourly measurements of 34 clinical signals, excluding static descriptors (*e. g.*, age, gender, unit type) and meta fields (*e. g.*, ICU length of stay) from the time series inputs. Static features are processed separately, with categorical variables (*e. g.*, gender) one-hot encoded into an extended static vector. Patients without any valid observations within the first 48 hours are excluded. Sepsis labels are assigned according to the official annotation, indicating whether any time point during the ICU stay is labeled as septic.

## C EXPERIMENT DETAILS

### C.1 BASELINES AND HYPERPARAMETERS

We evaluate our method against a wide range of baseline models, covering classical RNN variants, latent variable models, and recent graph- and transformer-based time series architectures. All baseline models are trained on the same data splits and evaluated using identical metrics to ensure a fair comparison.

For each model, we prioritize using the best-reported hyperparameter settings from original publications or official implementations. **Only when such configurations are unavailable or not directly transferable to our datasets**, we perform hyperparameter tuning within commonly used ranges. Specifically, the learning rate is selected from $\{1e^{-4}, 5e^{-4}, 1e^{-3}, 5e^{-3}, 1e^{-2}\}$, weight decay from

$\{1e^{-5}, 5e^{-5}, 1e^{-4}, 5e^{-4}, 1e^{-3}\}$, and batch size from $\{64, 128, 256, 512, 1024\}$ for PhysioNet (P12 and P19), and $\{8, 16, 32, 64, 128\}$ for MIMIC-III. **The following model-specific hyperparameter ranges are applied only in the absence of recommended configurations**.

**RNN-Mean**: RNN model where missing observations are imputed with global mean values (Che et al., 2018). Model-specific hyperparameters include the number of GRU hidden units selected from $\{32, 50, 64, 100\}$ and the latent dimension from $\{10, 20, 32, 64\}$.

**RNN-Decay**: RNN model that combines imputation of missing observations using an input decay mechanism (Che et al., 2018). Model-specific hyperparameters include the number of GRU hidden units selected from $\{32, 50, 64, 100\}$ and the latent dimension from $\{10, 20, 32, 64\}$.

**RNN-$\Delta_t$**: RNN model with input concatenated with the observation mask and sampling intervals. Model-specific hyperparameters include the number of GRU hidden units selected from $\{32, 50, 64, 100\}$ and the latent dimension from $\{10, 20, 32, 64\}$.

**GRU-D**: GRU model that applies a decaying mechanism in both input and hidden states (Che et al., 2018). Model-specific hyperparameters include the number of GRU hidden units selected from $\{32, 50, 64, 100\}$ and the latent dimension from $\{10, 20, 32, 64\}$.

**ODE-RNN**: RNN model that incorporates neural ODEs to model hidden states (Rubanova et al., 2019). Model-specific hyperparameters include the number of GRU hidden units from $\{50, 64, 100\}$, latent dimension from $\{20, 32, 64\}$, and set the number of layers in the ODE function network to $\{1, 2, 3\}$.

**L-ODE-RNN**: Latent ODE model with an RNN as the encoder (Chen et al., 2018). Model-specific hyperparameters include the latent dimension from $\{20, 32, 64\}$, encoder GRU hidden units from $\{50, 64, 100\}$, number of layers in the generative ODE function (`gen-layers`) from $\{1, 2, 3\}$.

**L-ODE-ODE**: Latent ODE model with ODE-RNN as the encoder (Rubanova et al., 2019). Model-specific hyperparameters include the encoder GRU hidden units from $\{50, 64, 100\}$, latent dimension from $\{20, 32, 64\}$, number of layers in both recognition (`rec-layers`) and generative (`gen-layers`) ODE functions from $\{1, 2, 3\}$.

**MTGNN**: Graph-based model for forecasting that jointly combines graph learning, graph convolution, and temporal convolution (Wu et al., 2020). Model-specific hyperparameters include the number of convolutional layers selected from $\{2, 3\}$ and the number of hidden channels from $\{32, 64, 128\}$.

**mTAND**: Utilizes an attention mechanism for continuous-time representations (Shukla & Marlin, 2021). Model-specific hyperparameters include the number of attention heads $H$ selected from $\{1, 2, 4\}$ and the GRU encoder hidden size from $\{20, 32, 64, 128\}$.

**Raindrop**: Graph-based model that handles irregular observations using a graph neural network (Zhang et al., 2022).

**Warpformer**: Learns time series at different scales with multiple warping and attention modules (Zhang et al., 2023). Model-specific hyperparameters include the dimensionality of hidden states $D$ selected from $\{32, 64\}$ and the number of attention layers $J$ selected from $\{2, 3\}$.

**MTSFormer**: Transformer-based model that utilizes time series data from four perspectives: Locality, Time, Spatio, and Irregularity (Zheng et al., 2024). Model-specific hyperparameters include the number of Transformer encoder layers selected from $\{2, 3, 4\}$ and the number of attention heads from $\{1, 2, 4\}$.

**RETAIN**: A reverse-time attention model originally designed for healthcare applications, which utilizes two-level attention mechanisms over input sequences (Choi et al., 2016). Model-specific hyperparameters include the hidden layer size $p$ selected from $\{64, 128, 256\}$ and embedding size $m$ selected from $\{64, 128, 256\}$.

**IMV-LSTM**: A model that captures temporal patterns and variable importance via a dual-stage attention mechanism within LSTM structures (Guo et al., 2019). Model-specific hyperparameters include the hidden layer size selected from $\{64, 128\}$.

**DA-RNN**: Dual-stage attention-based recurrent neural network that models both input features and temporal dependencies adaptively (Qin et al., 2017). Model-specific hyperparameters include the hidden layer size selected from $\{32, 64, 128\}$.

Table 11: Ablation study.

| Models | P12 | | P19 | |
|---|---|---|---|---|
| | AUROC | AUPRC | AUROC | AUPRC |
| GARLIC | 86.40±0.86 | 56.89±1.75 | 90.96±0.84 | 55.29±2.45 |
| w/o Missingness Indicator | 84.87±0.85 | 52.79±1.55 | 82.22±1.89 | 33.56±2.42 |
| w/o Decay Mechanism | 82.20±2.16 | 45.04±5.81 | 89.50±0.34 | 52.69±1.57 |
| w/o Signal-wise Encoder | 85.00±1.68 | 48.92±4.77 | 89.78±0.83 | 52.80±1.04 |
| w/o Signal-wise Attention | 74.67±4.88 | 36.70±2.92 | 89.30±0.83 | 51.98±1.84 |
| w/o Graph Message Passing | 84.67±0.77 | 49.45±2.63 | 88.39±0.78 | 51.66±2.62 |
| w/o GRU | 76.90±0.93 | 40.85±1.80 | 75.26±3.73 | 28.41±5.27 |
| w/o Alternating Decoupled Optimization | 85.32±1.81 | 54.95±3.79 | 90.08±0.58 | 54.36±1.44 |

## C.2 GARLIC HYPERPARAMETERS

We organize the hyperparameters of our proposed model into two categories: model architecture and training configuration. The following parameters are tuned during experimentation only when no recommended configuration is available.

For model architecture, the window size is selected from $\{2, 3, \ldots, 11\}$, which corresponds to time lags $\{1, 2, \ldots, 10\}$ used in the local temporal context, where the effective time lag $\tau =$ window size $-$ 1. The input feature dimension $D_f$ is chosen from $\{16, 32\}$, and the hidden state size $D_h$ for both encoder and decoder is selected from $\{128, 256, 512\}$.

For training, we tune the classification learning rate over $\{1e^{-5}, 5e^{-5}, 1e^{-4}\}$, classification weight decay from $\{1e^{-4}, 5e^{-4}, 1e^{-3}\}$, reconstruction learning rate from $\{1e^{-4}, 5e^{-4}, 1e^{-3}\}$, and reconstruction weight decay from $\{1e^{-5}, 5e^{-5}, 1e^{-4}, 5e^{-4}, 1e^{-3}\}$. The classification loss weight $\lambda_c$ is selected from $\{0.1, 1, 5, 10\}$, the graph regularization loss weight $\lambda_g$ is fixed at $0.01$, and the batch size is chosen from $\{64, 128, 256\}$.

# D ABLATION STUDIES AND SENSITIVITY ANALYSIS

## D.1 COMPONENT ABLATION

To better interpret the ablation results in table 11, we clarify the correspondence between each ablated variant and its associated module in GARLIC:

- **w/o Missingness Indicator**: Removes the missingness flag $m_{k,t}$ in the augmented input vector, as defined in equation 2. This prevents the model from distinguishing observed from imputed values.

- **w/o Decay Mechanism**: Disables the exponential decay-based imputation defined in equation 1, replacing time-aware estimation with static global means.

- **w/o Signal-wise Encoder**: Replaces the signal-specific encoders $\mathrm{MLP}_k$ in equation 3 with a shared encoder, thus ignoring the signal-specific heterogeneity in feature distributions.

- **w/o Signal-wise Attention**: Removes the cross-signal attention mechanism used at each timestep, as formulated in equation 6, thus aggregating signals uniformly.

- **w/o Graph Message Passing**: Eliminates the inter-signal message passing step in equation 5, disabling explicit modeling of relational dependencies among clinical variables.

- **w/o GRU**: Removes the GRU recurrence layer in equation 7, thereby discarding local temporal continuity modeling prior to temporal self-attention.

- **w/o Alternating Decoupled Optimization**: Trains the model jointly on reconstruction and classification objectives, rather than using the alternating optimization strategy described in section 3.4.

Table 12: Dataset statistics for the Human Activity dataset.

| Samples | Sensors | Class Distribution | Labels | Missing Ratio (%) | Task |
|---|---|---|---|---|---|
| 6442 | 12 | Walking: 1271, Falling: 1160, Lying: 896, Sitting: 2908, Standing up: 621, On all fours: 477, Sitting on the ground: 207 | 7 | $74.99 \pm 0.06$ | Posture Classification |

Table 11 quantifies the impact of disabling each core component of GARLIC on P12 and P19. Removing the missingness indicator reduces AUROC/AUPRC by 1.5/4.1 pts on P12 and 8.7/21.7 pts on P19, showing that flagging observed vs. imputed entries is critical under high missingness. Disabling the exponentialdecay mechanism costs 4.2/11.8 pts on P12 and 1.5/3.2 pts on P19, underscoring its role in capturing variablespecific dynamics. Replacing the persignal encoder with a uniform encoder/attention incurs moderate losses (1.4/7.9 pts on P12, 1.2/2.5 pts on P19), confirming the importance of modeling heterogeneity across clinical signals. Ablating signalwise attention causes a dramatic 11.7/20.2 pt drop on P12 (but minimal change on P19), reflecting datasetdependent sensitivity to local temporal encoding. Disabling graph message passing degrades performance by 1.7/7.4 pts on P12 and 2.6/6.6 pts on P19, confirming the benefit of modeling intersignal relations. Omitting the GRU yields the largest penalty (9.5/16.0 pts on P12, 15.7/26.9 pts on P19), highlighting its necessity for temporal continuity. Finally, reverting to joint endtoend training (no alternating optimization) leads to a modest 1.1/1.9 pt drop on P12 and 0.9/0.9 pt on P19, validating that decoupling reconstruction and classification stabilizes learning.

### D.2 HYPERPARAMETER SENSITIVITY FOR TIME LAG

Among the tunable hyperparameters, we focus our sensitivity analysis on the time lag $\tau$, which determines the length of the local temporal window used in both the attention and graph message passing modules. Unlike standard training hyperparameters (e.g., learning rate or batch size), $\tau$ is tightly coupled with the model's architectural design and directly affects how local temporal dependencies are captured. As such, it represents a model-specific architectural choice rather than a general training hyperparameter, and therefore warrants focused analysis.

Figure 5 shows the AUROC achieved on P12 and P19 for varying values of $\tau$. We observe that performance on P12 remains relatively stable across different $\tau$ values, while P19 exhibits a clearer performance peak at $\tau = 8$. This contrast highlights the dataset-dependent role of $\tau$: in longer sequences (P12), variations in $\tau$ affect a smaller part of the temporal context, resulting in a milder impact on performance. In shorter sequences (P19), the same variation significantly alters the model's view of input, leading to more noticeable performance differences.

Despite the general robustness of the model, the sensitivity patterns suggest that $\tau$ plays a non-negligible role in shaping temporal representations, especially in settings with limited context length. These findings justify treating $\tau$ as an architecture-level design choice that should be adapted to the characteristics of the dataset.

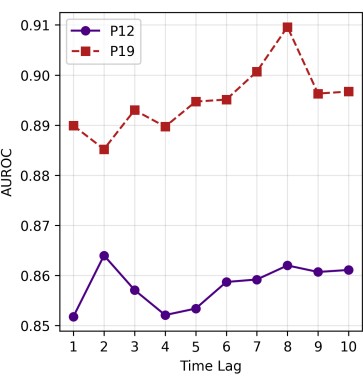

Figure 5: Sensitivity to $\tau$.

### E APPLICABILITY TO BROADER TASKS

Although many clinically relevant prediction problems–such as clinical deterioration prediction–are naturally framed as binary classification, we believe that assessing the model's performance on a broader set of tasks is crucial to fully demonstrate its versatility.

To this end, we extended our evaluation to two additional directions: multi-class classification and imputation. These experiments highlight GARLICs flexibility and generalization capability.

Table 13: Performance comparison on the Human Activity dataset.

| Model | Accuracy (%) | AUPRC (%) |
|---|---|---|
| GRU-D | $87.41 \pm 1.21$ | $81.29 \pm 2.83$ |
| ODE-RNN | $86.91 \pm 0.66$ | $77.11 \pm 1.88$ |
| L-ODE-ODE | $86.16 \pm 0.83$ | $74.63 \pm 3.74$ |
| L-ODE-RNN | $85.71 \pm 1.22$ | $71.04 \pm 1.53$ |
| mTAND | $88.72 \pm 1.46$ | $72.77 \pm 4.02$ |
| MTSFormer | $89.24 \pm 0.53$ | $79.14 \pm 1.46$ |
| WarpFormer | $89.28 \pm 0.87$ | $80.60 \pm 1.79$ |
| GARLIC | $\mathbf{91.21 \pm 1.03}$ | $\mathbf{84.70 \pm 1.92}$ |

## E.1 HUMAN ACTIVITY CLASSIFICATION

To evaluate GARLIC's performance on multi-class classification under irregular sampling, we conducted experiments on the Human Activity dataset, a widely used non-clinical benchmark for motion recognition. The dataset comprises multivariate time series collected from wearable sensors, encompassing 7 activity classes (see table 12). The task is a 7-class classification problem aimed at identifying the subjects activity from sensor readings over time. Following previous works (Rubanova et al., 2019; Shukla & Marlin, 2021), we applied standard preprocessing including normalization, sliding-window segmentation, and label alignment.

## E.2 BENCHMARKING ON PYRREGULAR

We also conducted additional experiments using the PYRREGULAR benchmark (Spinnato & Landi, 2025) and compared GARLIC with ROCKET under the same experimental settings. The results are summarized in table 14.

For the benchmark results, GARLIC demonstrates superior performance on the three healthcare datasets (P12, P19, MI3), confirming its strong suitability for the medical domain. For the Human Activity Recognition and sensor domains, GARLIC still achieves better performance on datasets such as LPA, CT, DD, DG, and DW, highlighting its broader applicability when signals exhibit meaningful interactions.

However, in the mobility domain (AN, AOC, GS, MP, SE, TA, VE), where the input signals are primarily GPS or accelerometer measurements, the signals do not possess strong interdependencies. As a result, GARLICs capability to leverage relationships among multiple signals cannot be fully utilized, leading to suboptimal performance. A similar limitation is observed on single-signal datasets (GM, GP, GX, GY, GZ, PGZ, SGZ, PL). Additionally, datasets like IW, which consist of frequency-domain data with minimal temporal correlations, are not well-suited for a model designed to capture temporal interactions across signals.

These results further highlight that GARLIC is particularly effective in scenarios where multiple signals are strongly correlated and exhibit rich temporal dependencies, as is typical in ICU medical data. This aligns with the motivation of our study: the medical domain presents uniquely challenging irregular multivariate time series, where capturing interactions among signals is crucial for accurate prediction. While GARLIC may be less advantageous in domains with weak or independent signals, its design and performance are well-suited to healthcare applications, justifying our focus on this domain.

## E.3 IMPUTATION

GARLICs modular architecture allows straightforward adaptation to tasks beyond classification. Specifically, the imputation task can be performed by disabling the third modulethe *Cross-Dimensional Sequential Attention*–and using only the first two components: the *Latent Feature Modeling* module and the *Time-Lagged Graph Message Passing* module. No further architectural changes are necessary.

Table 14: Comparison between ROCKET and GARLIC across datasets.

| Dataset | ROCKET | GARLIC | Sign |
|---|---|---|---|
| ABF | $0.17 \pm 0.00$ | $0.32 \pm 0.02$ | 1 |
| AN | $0.90 \pm 0.04$ | $0.86 \pm 0.01$ | 2 |
| AOC | $0.80 \pm 0.01$ | $0.64 \pm 0.05$ | 3 |
| APT | $0.96 \pm 0.00$ | $0.76 \pm 0.04$ | 3 |
| ARC | $0.99 \pm 0.01$ | $0.33 \pm 0.00$ | 3 |
| CT | $0.98 \pm 0.00$ | $0.98 \pm 0.00$ | 3 |
| DD | $0.54 \pm 0.03$ | $0.56 \pm 0.04$ | 1 |
| DG | $0.34 \pm 0.00$ | $0.75 \pm 0.02$ | 1 |
| DW | $0.42 \pm 0.00$ | $0.95 \pm 0.02$ | 1 |
| GM1 | $0.66 \pm 0.02$ | $0.49 \pm 0.05$ | 1 |
| GM2 | $0.57 \pm 0.05$ | $0.28 \pm 0.04$ | 1 |
| GM3 | $0.48 \pm 0.03$ | $0.22 \pm 0.09$ | 1 |
| GP1 | $0.89 \pm 0.02$ | $0.71 \pm 0.04$ | 1 |
| GP2 | $0.85 \pm 0.05$ | $0.65 \pm 0.05$ | 1 |
| GS | $0.31 \pm 0.15$ | — | 2 |
| GX | $0.70 \pm 0.01$ | $0.49 \pm 0.02$ | 1 |
| GY | $0.70 \pm 0.02$ | $0.46 \pm 0.02$ | 1 |
| GZ | $0.69 \pm 0.10$ | $0.36 \pm 0.04$ | 1 |
| IW | $0.53 \pm 0.00$ | $0.02 \pm 0.00$ | 200 |
| JV | $0.94 \pm 0.01$ | $0.97 \pm 0.00$ | 12 |
| LPA | $0.02 \pm 0.01$ | $0.49 \pm 0.05$ | 12 |
| MI3 | $0.35 \pm 0.00$ | $0.61 \pm 0.07$ | 17 |
| MP | $0.94 \pm 0.00$ | $0.92 \pm 0.01$ | 1 |
| P12 | $0.47 \pm 0.01$ | $0.68 \pm 0.01$ | 37 |
| P19 | $0.71 \pm 0.01$ | $0.77 \pm 0.02$ | 34 |
| PAM | $0.66 \pm 0.10$ | — | 52 |
| PGE | $0.40 \pm 0.00$ | $0.78 \pm 0.00$ | 9 |
| PGZ | $0.73 \pm 0.00$ | $0.70 \pm 0.04$ | 1 |
| PL | $0.85 \pm 0.01$ | $0.24 \pm 0.00$ | 1 |
| SAD | $0.98 \pm 0.00$ | $0.94 \pm 0.02$ | 13 |
| SE | $0.80 \pm 0.04$ | $0.69 \pm 0.12$ | 4 |
| SGZ | $0.88 \pm 0.02$ | $0.69 \pm 0.06$ | 1 |
| TA | $0.57 \pm 0.01$ | $0.77 \pm 0.01$ | 2 |
| VE | $0.94 \pm 0.02$ | $0.94 \pm 0.03$ | 2 |

To assess this capability, we carried out an imputation experiment on the P12 dataset, in which a fraction of the observations was randomly masked and used as ground truth. The model was trained to reconstruct missing values using mean squared error (MSE) as the loss function. We compared against several representative baselines supporting time series imputation, including L-ODE-RNN, L-ODE-ODE, and mTAND. The results are presented in table 15.

## F  LEARNED TIME-LAGGED SUMMARY GRAPH

To better understand how our model captures inter-signal dependencies for local reconstruction, we analyze the learned time-lagged summary graph $\mathcal{G}_\tau$ encoded by the adjacency matrix $\mathbf{W}_\tau$. While the primary goal of our model is outcome prediction, the graph module plays a crucial auxiliary role by modeling short-range temporal and cross-signal structure. Investigating the properties of the learned graph can thus shed light on the nature of the dependencies it captures.

We observe that the learned adjacency matrix $\mathbf{W}_\tau$ varies across different random seeds. This variability arises from the redundancy inherent in physiological signal dependencies and non-identifiable graph learning coupled with message passing (Yu et al., 2019; Compton et al., 2022). Since our model employs $\ell_1$ regularization (Lasso) to promote sparsity, it tends to select minimal subsets of edges sufficient for local reconstruction. Given the multitude of potential interactions among variables,

Table 15: Imputation performance on the P12 dataset under varying missing rates. Models were trained using MSE loss.

| Missing Rate | Model | MSE | MAE |
|---|---|---|---|
| 0.50 | L-ODE-RNN | $29.0073 \pm 0.4153$ | $21.3054 \pm 0.1788$ |
| | L-ODE-ODE | $27.0088 \pm 0.1868$ | $20.8137 \pm 0.1605$ |
| | mTAND | $\mathbf{12.3396 \pm 0.1397}$ | $\mathbf{10.2598 \pm 0.0677}$ |
| | GARLIC | $12.6943 \pm 0.4066$ | $10.6809 \pm 0.0992$ |
| 0.40 | L-ODE-RNN | $23.2620 \pm 0.5398$ | $17.0901 \pm 0.1745$ |
| | L-ODE-ODE | $21.5457 \pm 0.3561$ | $16.6419 \pm 0.1461$ |
| | mTAND | $\mathbf{9.3711 \pm 0.1347}$ | $\mathbf{8.0737 \pm 0.0720}$ |
| | GARLIC | $9.5636 \pm 0.1841$ | $8.1145 \pm 0.0940$ |
| 0.30 | L-ODE-RNN | $17.3302 \pm 0.5478$ | $12.8183 \pm 0.1454$ |
| | L-ODE-ODE | $16.0021 \pm 0.1276$ | $12.4869 \pm 0.0917$ |
| | mTAND | $6.9898 \pm 0.1088$ | $5.9852 \pm 0.0531$ |
| | GARLIC | $\mathbf{6.6576 \pm 0.1494}$ | $\mathbf{5.8061 \pm 0.0797}$ |
| 0.20 | L-ODE-RNN | $11.6055 \pm 0.2748$ | $8.5410 \pm 0.1130$ |
| | L-ODE-ODE | $10.7941 \pm 0.1971$ | $8.3196 \pm 0.0511$ |
| | mTAND | $4.5238 \pm 0.0718$ | $3.9589 \pm 0.0276$ |
| | GARLIC | $\mathbf{4.3584 \pm 0.2372}$ | $\mathbf{3.7085 \pm 0.0525}$ |
| 0.10 | L-ODE-RNN | $5.7913 \pm 0.1942$ | $4.3054 \pm 0.0448$ |
| | L-ODE-ODE | $5.3364 \pm 0.0626$ | $4.1734 \pm 0.0353$ |
| | mTAND | $2.2459 \pm 0.1128$ | $1.9778 \pm 0.0161$ |
| | GARLIC | $\mathbf{1.9737 \pm 0.1376}$ | $\mathbf{1.7834 \pm 0.0312}$ |

multiple sparse solutions can achieve comparable reconstruction performance. Consequently, each training run may converge to a different sparse instantiation from a set of approximately equivalent solutions. Importantly, we note that the learned graph $\mathcal{G}_\tau$ should be interpreted not as a recovery of the full physiological dependency network, but rather as a task-specific subgraph sufficient for local reconstruction. The graph is not supervised – it is not directly optimized using ground-truth labels, but instead emerges as an intermediate construct through the auxiliary reconstruction objective. As such, it acts as a functional tool to enhance representation learning, rather than a target of inference itself. The variability in learned edges across seeds thus reflects the presence of multiple valid subgraphs that support the same functional goal, rather than instability in model behavior.

To extract a more stable representation of inter-signal relationships, we compute the element-wise average of the learned adjacency matrices across five random seeds. Visualizations of this averaged graph are provided in figure 7, where the edge thickness reflects the mean weight learned. The resulting structure highlights consistent dependencies among physiological signals, capturing both intrinsic correlations and patterns indicative of external interventions. These observations suggest that our model effectively represents both endogenous signal dynamics and exogenous clinical actions that influence patient state.

To further assess the plausibility of the time-lagged summary graph, we conducted a post-hoc evaluation using five large language models (LLMs) as proxies for expert judgment, leveraging their encoded medical knowledge. Each edge was evaluated via a standardized prompt to ensure consistency (see Box F). While not a substitute for clinical validation, this provides a structured plausibility check.

Results indicate that four out of five LLMs judged over 80% of edges as reasonable. Notably, 9 edges were endorsed by all models and 3 edges by four models, demonstrating strong consistency. These findings align with established medical knowledge and support the plausibility of the learned graph.

Table 16: LLM-based post-hoc plausibility assessment of edges in the time-lagged summary graph. Each edge is categorized as Reasonable, Unclear, or Unreasonable.

| Model | Reasonable | Unclear | Unreasonable |
|---|---|---|---|
| Claude Opus 4 | 11 | 5 | 0 |
| Gemini 2.5 Pro | 14 | 2 | 0 |
| Grok 3 | 13 | 3 | 0 |
| ChatGPT O3 (Web Search) | 13 | 2 | 1 |
| DeepSeek (DeepThink R1, Search) | 13 | 2 | 1 |

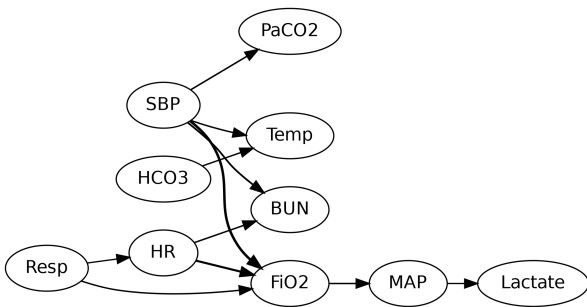

Figure 6: Visualization of learned graph after edge gating

Figure 7: Learned Summary Graph Visualization. Summary graph learned on dataset P19, averaged over 5 random seeds with a time lag of 8 hours.

---

**Prompt for LLM-based Plausibility Assessment**

You are a clinical expert with deep knowledge in human physiology, critical care, and multimodal patient monitoring. We are analyzing a time-lagged summary graph derived from ICU monitoring data over a 9-hour window. Each directed edge represents a potential influence – whether physiological, systemic, or intervention-based – between two variables. These influences may be direct or indirect, including multi-step pathways, systemic feedback, compensatory mechanisms, or clinical decisions. A variable does not need to be the sole or primary cause for its effect to be valid – edges may reflect partial, synergistic, or contributing influences within complex ICU dynamics.

Your task is to assess the plausibility of each edge as a causal or contributory effect, explicitly considering: - Direct physiological mechanisms - Indirect or multi-factorial pathways - Feedback loops and compensatory responses - Cascading effects of interventions and systemic derangements

For each edge, respond with: - **Reasonable** if plausible via any direct, indirect, partial, or synergistic mechanism - **Unclear** if the pathway is highly uncertain or not supported by known mechanisms - **Unreasonable** if it contradicts established physiological or clinical understanding

Provide a brief explanation (12 sentences), highlighting plausible direct or indirect routes, systemic interactions, or clinical reasoning.

**Edges to evaluate:** Resp $\to$ HR, Resp $\to$ FiO$_2$, HR $\to$ FiO$_2$, HR $\to$ MAP, FiO$_2$ $\to$ MAP, MAP $\to$ Lactate, SBP $\to$ HR, SBP $\to$ FiO$_2$, SBP $\to$ Temp, SBP $\to$ BUN, SBP $\to$ PaCO$_2$, HCO$_3^-$ $\to$ Temp, HCO$_3^-$ $\to$ BUN, Temp $\to$ BUN, Temp $\to$ FiO$_2$, BUN $\to$ FiO$_2$

---

**Future Work for Graph Learning:** To address non-identifiability inherent in sparse graph discovery, lightweight physiological priors such as known organ system linkages or treatment intervention markers can be incorporated to bias graph learning towards clinically plausible edges, and to explore consensus-based aggregation for more stable subgraphs. Extending $\mathcal{G}_\tau$ to a dynamic, window-free

setting can allow the adjacency weights to evolve continuously over time rather than be tied to fixed lags. Finally, static patient variables and multiscale time lags can also be integrated into a unified graph framework, so that both enduring patient traits and fast-changing physiological interactions can be captured in a single, interpretable structure.

## G    STATISTICAL TEST DETAILS

We employ two complementary statistical procedures to evaluate the interpretability of attribution scores under perturbation-based masking.

### G.1    PAGES L TEST FOR ORDERED ALTERNATIVES

Suppose we have $n$ repeated runs (blocks) and $k$ ordered conditions (here $k = 4$ for scenarios (c, d, b, a)). To test the strict ordering $c < d < b < a$, proceed as follows:

1. For each run $j = 1, \ldots, n$, rank the performance scores $k$ so that the smallest score receives rank 1 and the largest rank $k$. Denote the rank of condition $i$ in run $j$ by $r_{ij}$.

2. Compute the rank sums for each condition:

$$R_i = \sum_{j=1}^{n} r_{ij}, \quad i = 1, \ldots, k.$$

3. Assign weights $w_i = i$ corresponding to the hypothesized order $(c, d, b, a) \mapsto (1, 2, 3, 4)$.

4. Compute the Page statistic

$$L = \sum_{i=1}^{k} w_i R_i.$$

5. Under the null hypothesis of no ordering, $L$ can be compared to tabulated critical values (Page, 1963), or approximated by a normal distribution with

$$\mathbb{E}[L] = \frac{n\,k\,(k+1)^2}{4}, \quad \mathrm{Var}(L) \approx \frac{n\,k\,(k+1)(2k+1)}{24}\,\frac{(k+1)(2k+1)}{5},$$

yielding $Z = \frac{L - \mathbb{E}[L]}{\sqrt{\mathrm{Var}(L)}} \approx \mathcal{N}(0, 1)$.

6. Reject the null (and conclude a significant increasing trend $c < d < b < a$) if $L$ exceeds the critical value or if $Z > z_{1-\alpha}$.

### G.2    TWO ONESIDED TESTS (TOST) FOR EQUIVALENCE

To test whether two paired conditions (e. g. full vs. top50%) are practically equivalent within margin $\delta > 0$, we perform two onesided $t$-tests:

1. Compute the paired differences $d_j = a_j - b_j$ for $j = 1, \ldots, n$, their mean $\bar{d}$ and standard error $SE = s_d/\sqrt{n}$.

2. Test $H_{01} : \Delta \leq -\delta$ versus $H_{A1} : \Delta > -\delta$ by

$$t_1 = \frac{\bar{d} + \delta}{SE} \quad \text{with } n - 1 \text{ degrees of freedom,}$$

obtaining onesided $p_1 = 1 - F_{t_{n-1}}(t_1)$.

3. Test $H_{02} : \Delta \geq +\delta$ versus $H_{A2} : \Delta < +\delta$ by

$$t_2 = \frac{\bar{d} - \delta}{SE}, \quad p_2 = F_{t_{n-1}}(t_2).$$

4. The TOST $p$value is $\max(p_1, p_2)$. We declare equivalence (i.e. reject the union null) if both $p_1 < \alpha$ and $p_2 < \alpha$.

# H    ADDITIONAL CASE STUDIES

## H.1    SUPPLEMENTAL CASE STUDIES

To qualitatively assess the interpretability of our model, we select one correctly classified positive sample and one correctly classified negative sample from each of the P12 and P19 datasets. This yields four representative cases that allow us to examine the models attribution behavior under both high-risk and low-risk scenarios.

For each sample, we visualize the following components:

- **Importance Map of Observations (Raw):** This visualization displays the raw attribution scores assigned to each observed value. Observation values are normalized within each signal to enable consistent visual comparison across features. **The size of each marker reflects its corresponding importance score.**

- **Predicted Mortality Probability over Time:** We plot the model's predicted risk trajectory by feeding truncated input sequences of increasing length during inference, thereby revealing how prediction confidence evolves over time.

- **Importance per Signal:** We compute the aggregated importance scores for each signal by summing over all its observed time points, providing a global view of signal-level attribution.

- **Normalized Per-Signal Importance Map:** To reveal fine-grained temporal patterns, we normalize attribution scores within each signal channel. At this stage, we intentionally skip the redistribution step described in equation 11 and directly apply the attribution mask. This is because redistribution is only used to evenly reassign the importance originally attributed to imputed values back to the observed data within the same signal. Since our goal here is to examine the temporal structure of observed inputs, omitting redistribution helps avoid diluting the attribution signal and provides a clearer view of temporal dynamics. **In the resulting visualization, color intensity represents importance: red indicates high attribution, while blue denotes relatively low importance.**

All visualizations include only signals with observed values; signals with no observations are omitted for clarity.

**P12 Dataset Signals.** Albumin (g/dL), ALP (IU/L), ALT (IU/L), AST (IU/L), Bilirubin (mg/dL), BUN (mg/dL), Cholesterol (mg/dL), Creatinine (mg/dL), DiasABP (mmHg), FiO$_2$ (0–1), GCS (score), Glucose (mg/dL), HCO$_3$ (mmol/L), HCT (%), HR (bpm), K (mEq/L), Lactate (mmol/L), Mg (mmol/L), MAP (mmHg), MechVent (0/1), Na (mEq/L), NIDiasABP (mmHg), NIMAP (mmHg), NISysABP (mmHg), PaCO$_2$ (mmHg), PaO$_2$ (mmHg), pH (0–14), Platelets (cells/nL), RespRate (bpm), SaO$_2$ (%), SysABP (mmHg), Temp (°C), TropI ($\mu$g/L), TropT ($\mu$g/L), Urine (mL), WBC (cells/nL).

**P19 Dataset Signals.** HR (bpm), O2Sat (%), Temp (°C), SBP (mmHg), MAP (mmHg), DBP (mmHg), Resp (breaths/min), EtCO2 (mmHg), BaseExcess (mmol/L), HCO$_3$ (mmol/L), FiO$_2$ (%), pH, PaCO$_2$ (mmHg), SaO$_2$ (%), AST (IU/L), BUN (mg/dL), Alkalinephos (IU/L), Calcium (mg/dL), Chloride (mmol/L), Creatinine (mg/dL), Bilirubin_direct (mg/dL), Glucose (mg/dL), Lactate (mg/dL), Magnesium (mmol/dL), Phosphate (mg/dL), Potassium (mmol/L), Bilirubin_total (mg/dL), TroponinI (ng/mL), Hct (%), Hgb (g/dL), PTT (s), WBC ($10^3/\mu$L), Fibrinogen (mg/dL), Platelets ($10^3/\mu$L).

## H.2    DISCUSSION

Our case study reveals nuanced insights into how the model assigns and utilizes feature importance across different granularity levels, offering a faithful interpretation of its decision-making process in clinical time-series data.

**Global-Level (Raw) Importance Attribution.** We observe that signals with fewer observations tend to receive disproportionately higher per-observation importance scores. This reflects a natural behavior: when a channel is sparsely sampled, each observation contributes a larger fraction of the total available information for that signal, prompting the model to rely more heavily on these sparse events. In positive samples (Sample02 and Sample04), these sparse observations frequently coincide with sharp increases in prediction probability, indicating their critical role in outcome prediction.

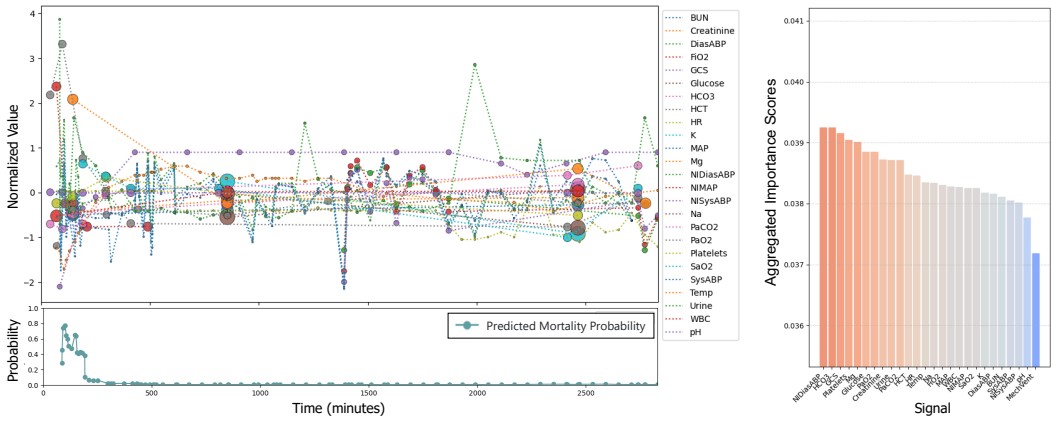

(a) Importance Map of Observations (Raw) / Predicted Mortality Probability  (b) Importance per Signal

Figure 8: Sample 01: Surviving patient from P12.

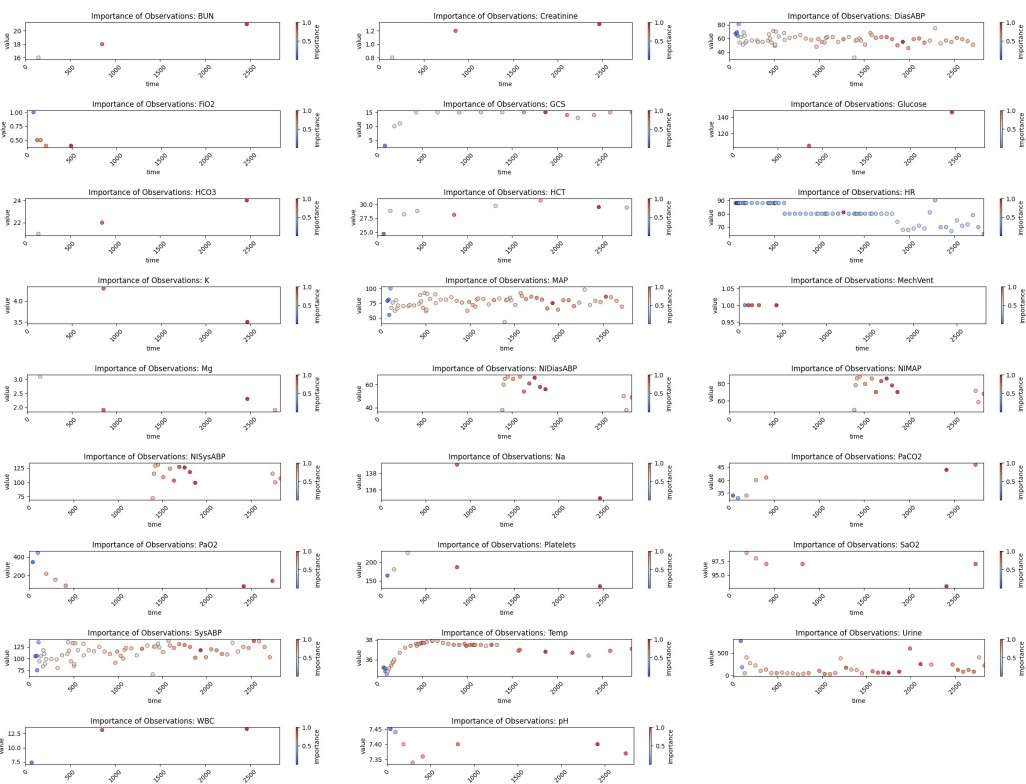

Figure 9: Sample 01: Normalized per-signal importance map.

Conversely, in negative samples (Sample01 and Sample03), similar observations occur during stable, low-risk phases and do not cause prediction spikes—highlighting their role in characterizing patient stability. These patterns suggest the model is not merely reactive to anomalies, but rather context-aware in distinguishing between deterioration and normalcy.

**Signal-Level Importance Distribution.** When aggregating importance scores across each signal, we find relatively small variance (within 5%—10%), indicating that the model does not rely excessively on a few dominant signals. Instead, it integrates information across multiple signals in a synergistic manner. This likely reflects two key modeling traits: (1) a preference for global pattern recognition

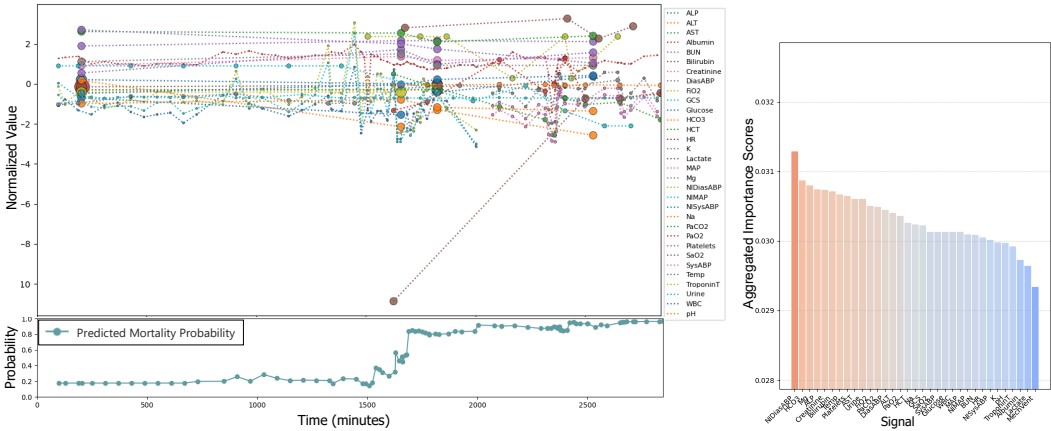

(a) Importance Map of Observations (Raw) / Predicted Mortality Probability    (b) Importance per Signal

Figure 10: Sample 02: Non-surviving patient from P12.

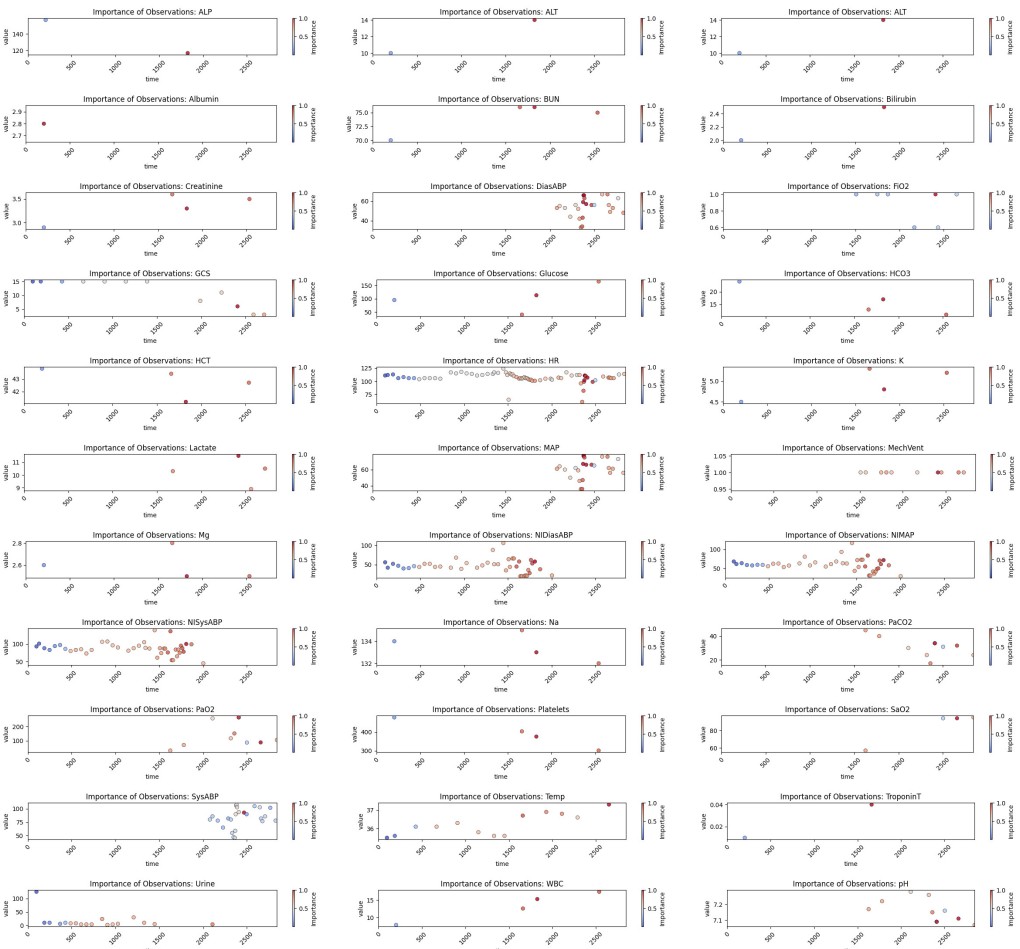

Figure 11: Sample 02: Normalized per-signal importance map.

rather than localized feature triggers; and (2) limited granularity in distinguishing signal-specific importance, possibly due to a generalization bias to avoid overfitting to any single input source. Such behavior is desirable in complex clinical settings, where predictive stability is favored over fragile saliency.

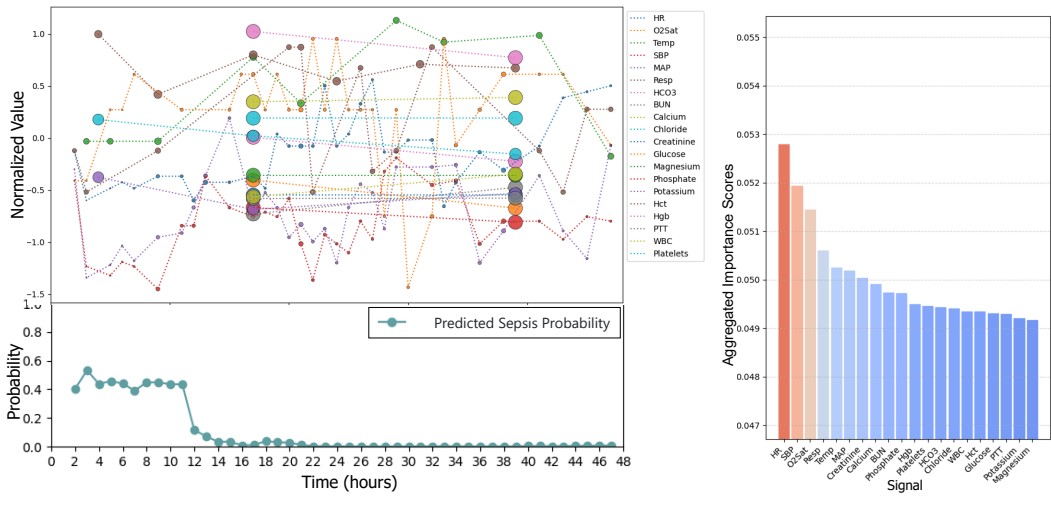

(a) Importance Map of Observations (Raw) / Predicted Sepsis Probability    (b) Importance per Signal

Figure 12: Sample 03: Non-sepsis patient from P19.

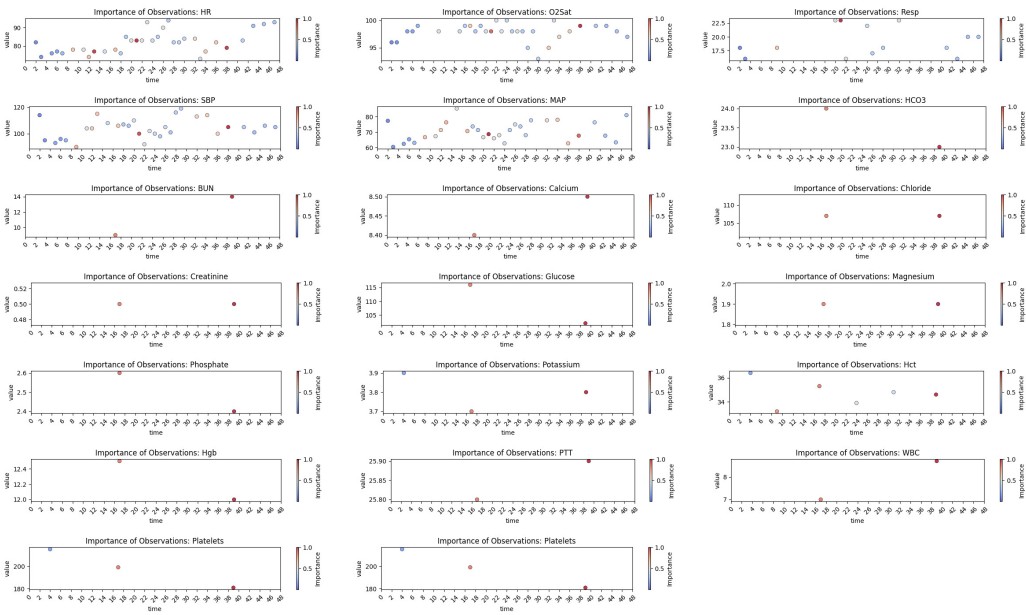

Figure 13: Sample 03: Normalized per-signal importance map.

**Temporal-Level Sensitivity (Normalized Importance).** After normalizing importance at the signal level, we find the model exhibits strong temporal sensitivity to two specific types of patterns. First, it reacts strongly to high-variation events—abrupt changes often corresponding to acute physiological deterioration. For instance, in Sample02, the model highlights spikes in HR, MAP, and DiasABP, as well as increasing fluctuations in NISysABP, NIDiasABP, and NIMAP. Second, the model emphasizes abnormal value onsets, such as HR > 90, temperature > 38°C, and WBC < 4 in Sample04—markers that are clinically associated with onset of sepsis. These behaviors indicate the model is sensitive to early signs of clinical instability, validating its prioritization from a physiological standpoint.

**Temporal Alignment with Clinical Semantics.** Interestingly, we find that the temporal distribution of feature importance naturally aligns with the models prediction direction, revealing contrasting attribution patterns for different outcomes. For example, in the GCS channel, the model assigns greater importance to rising trends and values above 8 in Sample01 (predicted as a survivor), whereas

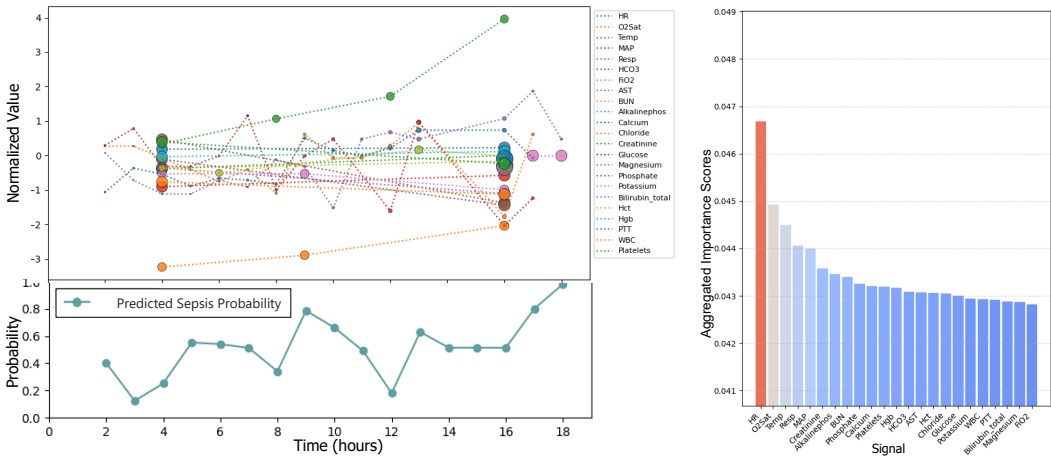

(a) Importance Map of Observations (Raw) / Predicted Sepsis Probability    (b) Importance per Signal

Figure 14: Sample 04: Sepsis patient from P19.

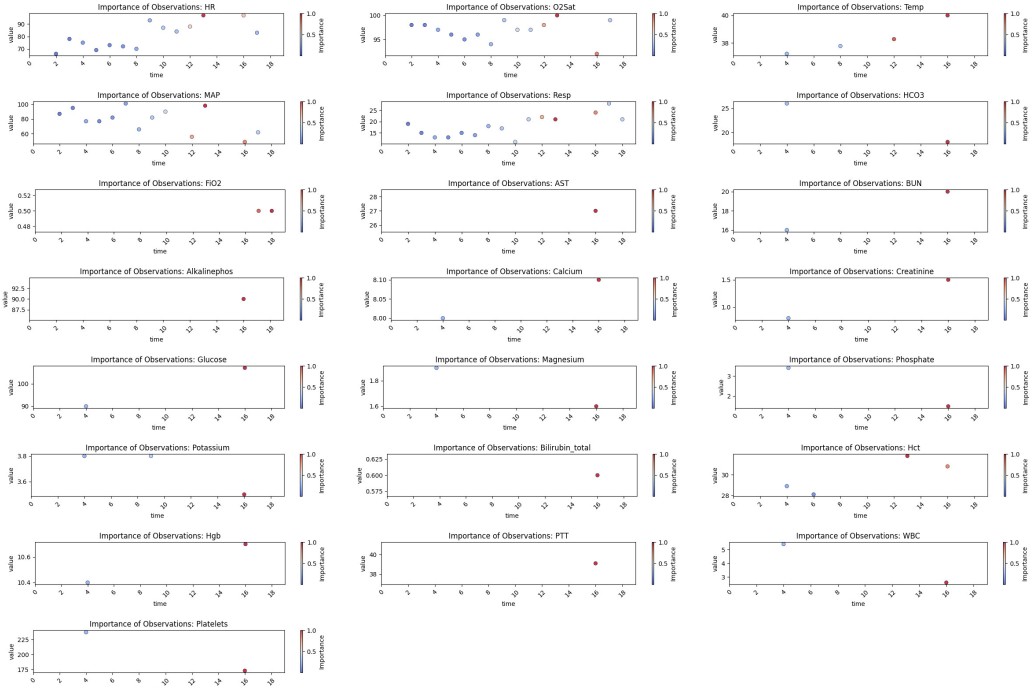

Figure 15: Sample 04: Normalized per-signal importance map.

it highlights declining trends and values below 8 in Sample02 (predicted as a non-survivor). This indicates that the model does not simply react to specific values or patterns in isolation, but rather emphasizes temporal segments that are most indicative of the predicted class. Similar patterns are observed in Sample01 for FiO$_2$ and temperature, where segments following a transition into normal ranges receive higher importance compared to those in abnormal ranges—suggesting that the model focuses on evidence of physiological recovery when predicting survival. These attribution behaviors also align well with the dynamics of predicted probabilities, as periods of high attribution typically coincide with sharp probability shifts, particularly in positive cases. Collectively, these observations demonstrate that the model assigns importance in a way that is both temporally grounded and semantically consistent with its outcome predictions.

**Limitations and Potential Improvements of Interpretability.** While many of GARLICs attributions align with clinical intuition, there are instances where high importance is assigned to physiologically stable measurements, raising questions about interpretability. Specifically, this may occur due to noise or instability in the attribution process, or because GARLIC captures statistically correlated patterns that lack clear clinical meaning, often serving as indirect proxies for latent-space reconstruction or temporal encoding rather than directly predicting outcomes.

However, this behavior is mainly observed in true-negative (healthy) samples, such as Sample 03 in figure 13, where highlighting stable values is physiologically reasonable. For example, GARLIC emphasizes measurements around hour 21 in HR, SBP, MAP, O2Sat, and Resp. Although it is unclear why these particular timepoints are highlighted over similar stable points, stable physiological values in non-sepsis cases provide evidence against clinical deterioration, suggesting that such attributions are both reasonable and clinically meaningful.

To improve the clinical consistency of GARLICs interpretations, we propose three complementary strategies: **Clinical Priors Integration:** Incorporate expert-defined physiological ranges (e.g., normal MAP or WBC intervals) via an attribution-regularization loss, which penalizes high importance assigned to normal values and emphasizes clinically significant anomalies. *Applicability:* When robust clinical priors are available. *Limitation:* Effectiveness depends on prior quality and may miss rare pathological events or complex interactions outside standard ranges. **Stability-Aware Attribution Filtering:** Apply temporal smoothness constraints to attention weights to reduce isolated spikes and emphasize coherent trends. *Applicability:* When signals are densely sampled and temporal continuity is important. *Limitation:* May obscure brief but clinically relevant deviations; not suitable for sparse data. **Gradient-Based Attention Exploration:** Augment attention with gradient-based saliency methods (e.g., Grad-CAM-style) to capture fine-grained dependencies (Leem & Seo, 2024; Barkan et al., 2021). *Applicability:* When detailed feature interactions are critical and sufficient computational resources are available. *Limitation:* Sensitive to noise in high-dimensional data and introduces additional computational overhead.

In summary, GARLIC provides a foundational attribution mechanism for clinical time series, capable of capturing both risk-elevating and stabilizing patterns across signals and time. Our case studies show that the model attends to clinically meaningful events, differentiates attribution behaviors across predicted outcomes, and distributes importance in a way that largely aligns with clinical intuition. To further enhance interpretability and reliability, we propose three complementary extensionsclinical priors integration, stability-aware filtering, and gradient-based saliencywhich offer flexible paths to adapt the attributions based on signal characteristics and computational constraints. While some attribution inconsistencies remain, these analyses demonstrate the potential of structured temporal attributions to provide actionable insights and reinforce trust in clinical decision-making systems.

