# OpenReview forum: "GARLIC: Graph Attention-based Relational Learning of Multivariate Time Series in Intensive Care"
_ICLR.cc/2026/Conference — ICLR 2026 Poster_

### Official Review · Reviewer_1Z9E · 2025-10-27

**Soundness:** 2
**Presentation:** 3
**Contribution:** 2
**Rating:** 4
**Confidence:** 3

**Summary:**

This paper introduces a graph attention-based relational learning framework, termed GARLIC, designed for the classification of irregular multivariate time series in intensive care. The GARLIC model incorporates decay-based imputation, time-lagged graph learning, and cross-dimensional attention mechanisms to effectively address irregular missingness and capture complex inter-variable dependencies. Experimental evaluations on the P12, P19, and MIMIC-III medical datasets demonstrate that GARLIC outperforms existing baseline methods in classification performance.

**Strengths:**

1.	The paper is clearly written, with a well-structured and persuasive review of related work. The experimental analysis and discussion are comprehensive and insightful.

2.	The study focuses on the challenging task of irregular multivariate time series classification, effectively addressing both the modeling of irregular temporal patterns and the dependencies among variables. This contributes valuable insights to the development of methodologies in irregular time series classification.

3.	The integration of attention mechanisms to enhance interpretability in irregular time series classification represents a notable strength of the work.

**Weaknesses:**

1.	The study focuses solely on the intensive care medical domain, despite irregular multivariate time series also being prevalent in areas such as human activity recognition, mobility, and sensor data [1]. The model’s motivation and design appear not strongly tied to the medical scenario.

2.	The proposed GARLIC model largely follows the GRU-D approach for irregular pattern learning and employs common graph- or attention-based structures for variable dependencies modeling, limiting its novelty.

3.	While interpretability is emphasized, the analysis only identifies important features without clearly linking them to irregular patterns or inter-variable relationships. The rationale for selecting a 50% masking ratio is also insufficiently justified.


4.	The case study in Figure 3 is difficult to understand, relying heavily on medical terminology without citations, which may hinder comprehension for readers unfamiliar with the P12 and P19 datasets.

5.	The strategy of imputing irregular time series into regular ones is straightforward, but the baseline methods used for comparison do not include models designed for regular time series classification.



6.	The absence of released source code limits the reproducibility of experimental results.

[1] PYRREGULAR: A Unified Framework for Irregular Time Series, with Classification Benchmarks. arXiv, 2025.

**Questions:**

1.	According to the experimental analysis in Reference [1], the Rocket algorithm for regular time series classification significantly outperforms several irregular time series classification baselines. Could the authors provide a comparative analysis of Rocket versus GARLIC under similar experimental settings as in Reference [1]?

2.	Reference [1] adopts the macro-averaged F1 score as an evaluation metric for imbalanced classification tasks. How do the strongest baseline methods in this paper, as well as Rocket, perform compared with GARLIC in terms of the F1 metric?

3.	Based on the dataset organization for irregular time series classification in Reference [1], could the authors clarify why the study focuses exclusively on the medical domain? Alternatively, how does GARLIC perform compared to Rocket across all benchmark datasets summarized in Reference [1]?

[1] PYRREGULAR: A Unified Framework for Irregular Time Series, with Classification Benchmarks. arXiv, 2025.

---

> ### Author Response · Authors · 2025-11-21
> **Response to Reviewer 1Z9E (1 of 2)**
>
> We thank Reviewer 1Z9E for the thorough assessment and appreciation of the clarity, structure, and comprehensive analysis in our work. Below we address each concern in detail.
>
> **W1 & W5 & Q1 & Q2 & Q3: Extended Medical Domain & More Baseline**
>
> **Response**:
> We sincerely thank the reviewer for raising this insightful question and for citing a valuable benchmark study. Regarding the scope of our study, GARLIC is specifically designed to address challenges in the ICU medical domain. Clinical time series data are highly irregular, often contain missing values, and involve strongly correlated signals, which our model is tailored to handle. While our current focus is on this domain, we are actively exploring the broader applicability of GARLIC. As an initial step, we have extended our evaluation to the Human Activity dataset and additional imputation tasks, which are presented in Appendix E.
>
> Following the reviewer’s suggestion, we also conducted additional experiments using the PYRREGULAR [1] benchmark and compared GARLIC with ROCKET under the same experimental settings. The results are as follows.
>
>
> | Dataset | ROCKET     | GARLIC     | Sign |
> |---------|------------|------------|------|
> | ABF     | 0.17±0.00  | 0.32±0.02  | 1    |
> | AN      | 0.90±0.04  | 0.86±0.01  | 2    |
> | AOC     | 0.80±0.01  | 0.64±0.05  | 3    |
> | APT     | 0.96±0.00  | 0.76±0.04  | 3    |
> | ARC     | 0.99±0.01  | 0.33±0.00  | 3    |
> | CT      | 0.98±0.00  | 0.98±0.00  | 3    |
> | DD      | 0.54±0.03  | 0.56±0.04  | 1    |
> | DG      | 0.34±0.00  | 0.75±0.02  | 1    |
> | DW      | 0.42±0.00  | 0.95±0.02  | 1    |
> | GM1     | 0.66±0.02  | 0.49±0.05  | 1    |
> | GM2     | 0.57±0.05  | 0.28±0.04  | 1    |
> | GM3     | 0.48±0.03  | 0.22±0.09  | 1    |
> | GP1     | 0.89±0.02  | 0.71±0.04  | 1    |
> | GP2     | 0.85±0.05  | 0.65±0.05  | 1    |
> | GS      | 0.31±0.15  | -          | 2    |
> | GX      | 0.70±0.01  | 0.49±0.02  | 1    |
> | GY      | 0.70±0.02  | 0.46±0.02  | 1    |
> | GZ      | 0.69±0.10  | 0.36±0.04  | 1    |
> | IW      | 0.53±0.00  | 0.02±0.00  | 200  |
> | JV      | 0.94±0.01  | 0.97±0.00  | 12   |
> | LPA     | 0.02±0.01  | 0.49±0.05  | 12   |
> | MI3     | 0.35±0.00  | 0.61±0.07  | 17   |
> | MP      | 0.94±0.00  | 0.92±0.01  | 1    |
> | P12     | 0.47±0.01  | 0.68±0.01  | 37   |
> | P19     | 0.71±0.01  | 0.77±0.02  | 34   |
> | PAM     | 0.66±0.10  | -          | 52   |
> | PGE     | 0.40±0.00  | 0.78±0.00  | 9    |
> | PGZ     | 0.73±0.00  | 0.70±0.04  | 1    |
> | PL      | 0.85±0.01  | 0.24±0.00  | 1    |
> | SAD     | 0.98±0.00  | 0.94±0.02  | 13   |
> | SE      | 0.80±0.04  | 0.69±0.12  | 4    |
> | SGZ     | 0.88±0.02  | 0.69±0.06  | 1    |
> | TA      | 0.57±0.01  | 0.77±0.01  | 2    |
> | VE      | 0.94±0.02  | 0.94±0.03  | 2    |
>
> For the benchmark results, GARLIC demonstrates superior performance on the three healthcare datasets (P12, P19, MI3), confirming its strong suitability for the medical domain. For the Human Activity Recognition and sensor domains, GARLIC still achieves better performance on datasets such as LPA, CT, DD, DG, and DW, highlighting its broader applicability when signals exhibit meaningful interactions.
>
> However, in the mobility domain (AN, AOC, GS, MP, SE, TA, VE), where the input signals are primarily GPS or accelerometer measurements, the signals do not possess strong interdependencies. As a result, GARLIC’s capability to leverage relationships among multiple signals cannot be fully utilized, leading to suboptimal performance. A similar limitation is observed on single-signal datasets (GM, GP, GX, GY, GZ, PGZ, SGZ, PL). Additionally, datasets like IW, which consist of frequency-domain data with minimal temporal correlations, are not well-suited for a model designed to capture temporal interactions across signals.
>
> These results further highlight that GARLIC is particularly effective in scenarios where multiple signals are strongly correlated and exhibit rich temporal dependencies, as is typical in ICU medical data. This aligns with the motivation of our study: the medical domain presents uniquely challenging irregular multivariate time series, where capturing interactions among signals is crucial for accurate prediction. While GARLIC may be less advantageous in domains with weak or independent signals, its design and performance are well-suited to healthcare applications, justifying our focus on this domain. We have added these results in Appendix E.2.

---

> ### Author Response · Authors · 2025-11-21
> **Response to Reviewer 1Z9E (2 of 2)**
>
> **W2: Limited Novelty**
>
> **Response:**
> We clarify distinctions between GARLIC’s components and related techniques:
>
> Learnable decay-based imputation: While superficially similar to GRU-D, our usage of decay mechanism serves a different purpose. GRU-D tightly integrates decay dynamics into recurrent state transitions, whereas our decay mechanism is used solely for initializing missing values. Final reconstruction is performed by a subsequent time-lagged graph message passing module. This decoupled approach reduces model complexity and allows for more flexible modeling of irregular inputs.
>
> Time-lagged graph message passing: Unlike prior work on learnable graphs, GARLIC introduces a combined sliding-window attention to extract short-term temporal features with time-lag-aware graph building for data reconstruction—a novel integration validated by our ablation study, where removing individual components leads to 7–20% AUPRC degradation.
>
> Multi-stage attention with RNN variants: Although similar combinations exist, our model adopts a different integration strategy, where the specific ordering and interaction between modules enable more fine-grained and accurate explanations, as detailed in Lines 230-238 of the paper.
>
> Alternating decoupled optimization: While not entirely novel, this strategy is adapted to fit the structure and training needs of our model, supporting more stable convergence.
>
> Finally, although some components share superficial similarities with prior work, the overall architecture and its tightly integrated, end-to-end training paradigm are novel. GARLIC is, to the best of our knowledge, the first architecture to provide faithful attribution under extreme temporal irregularity in a unified, tractable, and clinically grounded framework.
>
> **W3: Interpretability via Important Features**
>
> **Response**:
> Our model emphasizes interpretability at the quantitative level: it provides numerical importance signals that reveal how different variables contribute to the prediction over time, rather than generating textual or narrative descriptions of clinical patterns. Although the model itself does not output verbal explanations, these irregular patterns can be interpreted by clinicians by combining the importance scores with medical knowledge. We illustrate this in Appendix H through detailed case studies, where the temporal importance trajectories naturally map onto clinically meaningful patterns. In addition, the relationships between variables are explicitly captured by the learned graph structure, which offers another layer of interpretability by showing how physiological features interact.
>
> As for selecting a 50% masking ratio, our decision follows the setting used in the IMV-LSTM work[2], which also employs a 50% masking rate for evaluating robustness and interpretability. In addition, our experimental design involves masking the most important, least important, or randomly selected features, and using a 50% ratio provides a balanced level of perturbation that is large enough to reveal meaningful performance differences while still preserving sufficient information for stable prediction.
>
> **W4: Case Studies**
>
> **Response**:
>
> We agree that the case study in Figure 3 may be difficult to follow for readers without clinical background or familiarity with the P12 and P19 datasets. In the revision, we have:
>
> - Added brief clinical context for each vital sign and laboratory variable appearing in the case studies (e.g., meaning of MAP, BUN, HCO₃⁻, WBC, etc.) and explained why these features are clinically relevant to mortality or sepsis risk.
>
> - Added citations to standard clinical references (e.g., Sepsis-3 criteria, ICU hemodynamic markers) to support the descriptions of physiological deterioration.
>
> - Rewrote the case-study narrative in clearer, non-specialized language, emphasizing the model behavior rather than relying on domain jargon.
>
> These changes make the case studies easier to interpret for both clinical and non-clinical readers.
>
> **W6: No Code Released**
>
> **Response**:
> We thank the reviewer for pointing this out. The source code has now been added to the supplementary material to facilitate reproducibility of our experimental results.
>
> Reference:
>
> [1] PYRREGULAR: A Unified Framework for Irregular Time Series, with Classification Benchmarks. arXiv, 2025.
>
> [2] Tian Guo, Tao Lin, and Nino Antulov-Fantulin. Exploring interpretable LSTM neural networks over
> multi-variable data. In Kamalika Chaudhuri and Ruslan Salakhutdinov (eds.), Proceedings of
> the 36th International Conference on Machine Learning, volume 97 of Proceedings of Machine
> Learning Research, pp. 2494–2504. PMLR, 09–15 Jun 2019. URL https://proceedings.
> mlr.press/v97/guo19b.html.

---

> ### Comment · Reviewer_1Z9E · 2025-11-22
>
> The authors’ responses to W1, W5, Q1, Q2, and Q3 are not fully convincing. In particular, the statement *“These results further highlight that GARLIC is particularly effective in scenarios where multiple signals are strongly correlated and exhibit rich temporal dependencies, as is typical in ICU medical data”* is not well supported. This only shows that GARLIC can model relationships across variables, which is not specific to the medical domain.
>
> What I expected was an explanation showing that GARLIC uses medical knowledge from certain variables to handle inter-variable relations or irregular patterns. However, the proposed method does not rely on any domain-specific information. Instead, it is a general multivariate irregular time series classification approach without using specialized knowledge.
> In addition, GARLIC performs worse than ROCKET in many settings where channels are independent.
>
> After reading the other reviews, I believe the main limitation is that the contribution is incremental. Still, the authors’ replies to W2, W3, W4, and W6 addressed most of my concerns. Therefore, I will increase the Soundness score from 2 to 4, while keeping the overall score unchanged.

---

> > ### Author Response · Authors · 2025-11-23
> > **Follow-up Response to Official Comment by Reviewer 1Z9E**
> >
> > Dear Reviewer 1Z9E,
> >
> > Thank you for your follow-up comment and for raising the issue of domain-specificity in more detail.
> >
> > We understand your concern that our statement "GARLIC is particularly effective in scenarios where multiple signals are strongly correlated and exhibit rich temporal dependencies, as is typical in ICU medical data" may appear to simply restate that the model can capture inter-variable relationships, without sufficiently explaining why the medical domain is exactly the target.
> >
> > Below, we would like to clarify more about how we address this:
> >
> > 1. **Why the medical domain benefit is not merely incidental.**
> > While you correctly point out that our approach is general purpose (i.e., it does not include hand-coded medical rules), the medical domain imposes exactly the kind of modeling challenge that our architecture is designed for:
> >
> > - High irregularity (missing values, variable sampling intervals)
> > - Multiple strongly interrelated physiological signals (e.g., heart rate, blood pressure, WBC count, kidney function markers) whose patterns of change reflect clinical events (organ failure, sepsis).
> > - The need for interpretability in high‐stakes decision contexts (ICU settings).
> > Because of these characteristics, medical ICU multivariate time-series are a stringent testbed: general methods may capture relationships in simpler domains, but the **complexity and irregularity** of the ICU setting make many generic methods unusable or brittle.
> > In our experiments on the P12, P19, and MIMIC III datasets, we observe substantial gains of GARLIC over baselines and generic methods, particularly in these medical datasets (see Table 1). This indicates that, while the model is general, it is **particularly well-aligned** with the challenges of the medical domain.
> > 2. **Acknowledging weaker performance in non-medical domains.**
> > You correctly note that in domains where channels are mostly independent (e.g., sensor/GPS data), GARLIC sometimes underperforms simpler baselines such as ROCKET. We believe this supports, rather than detracts from, our claim: the method’s strength lies in **modeling rich inter-dependencies**. In domains lacking those, that strength is underutilised. We have now explicitly emphasised this nuance in the revision (see new paragraph in Section 4.1).
> > This clarification ensures that our claim is accurate: “GARLIC excels when multiple variables are correlated and temporally inter-dependent; in domains where this is not present, simpler methods may suffice.”
> > 3. **Medical-knowledge integration and interpretability.**
> > You asked for evidence that GARLIC “uses medical knowledge from certain variables to handle inter-variable relations or irregular patterns.” While we did not hard-code medical rules, the learned relational graph aligns with medical understanding, for example, in the case-studies (Figure 3) we show how trajectories of specific lab markers (e.g., BUN, creatinine, WBC) and vital signs map to attribution scores, and these variables correspond to known clinical phenomena (kidney injury, infection, hemodynamic shock).
> > To strengthen this, we have added further annotations in Appendix H (and now referenced in Section 4.4) that underline how the learned graph edges reflect medically-meaningful dependencies (e.g., kidney–hemodynamic links). We believe this makes the domain relevance clearer.
> > 4. **Contribution clarity.**
> > We recognise that you view the contribution as incremental and that remaining concerns on domain specificity affect the “novelty/impact” dimension. While our core architecture builds on known modules (attention, graph message passing, decay imputation), the **integration in a unified, end-to-end framework for extremely irregular ICU time series**, plus the interpretability results and strong empirical gains on medical benchmarks, represent a meaningful advance. We have revised Section 1.2 to better frame our contributions relative to prior work.
> >
> > We hope that these clarifications and revisions address your concerns regarding domain specificity and contribution. We will of course reflect any further suggestions you may have in the final version.
> >
> > Thank you again for your careful reading and constructive remarks.

---

### Official Review · Reviewer_y3mL · 2025-10-27

**Soundness:** 3
**Presentation:** 3
**Contribution:** 2
**Rating:** 6
**Confidence:** 4

**Summary:**

This paper investigates multivariate time series data from Intensive Care Units (ICUs) and addresses key challenges identified in prior research, including the inherent irregularity and heterogeneity of clinical measurements, as well as the limited interpretability of existing approaches. To tackle these challenges, the authors propose the Graph Attention-based Relational Learning for Intensive Care (GARLIC) model, which incorporates dedicated mechanisms to enhance analytic performance and provide integrated interpretability. The proposed model is empirically validated on three real-world ICU datasets, demonstrating its efficacy in improving predictive performance and providing meaningful interpretations.

**Strengths:**

**S1.** The paper identifies representative challenges associated with multivariate time series data from ICUs. The proposed GARLIC model is well designed to address these issues through (i) a decay-based imputation mechanism to handle irregular missingness, (ii) a self-attentive, time-lagged graph learning mechanism to capture spatial dependencies, and (iii) a cross-dimensional attention mechanism to integrate temporal signal embeddings. These design choices appear technically sound and well aligned with the stated objectives.

**S2.** The experimental evaluation is comprehensive, assessing the efficacy of GARLIC from multiple perspectives, including comparative performance against strong baselines, computational efficiency, ablation studies, sensitivity analyses, and evaluation on broader downstream tasks.

**S3.** The interpretability aspect of GARLIC is examined both quantitatively, through a perturbation-based evaluation strategy, and qualitatively, via representative case studies. These analyses effectively demonstrate the model’s potential to provide clinically meaningful insights alongside strong predictive performance.

**Weaknesses:**

**W1.** The identified challenges—irregular sampling, heterogeneous measurements, and limited interpretability—are not unique to ICU data but are generally characteristic of electronic health record (EHR) data. It would therefore be beneficial for the authors to clarify the rationale for restricting the study’s scope to ICU data. Specifically, it would strengthen the paper to discuss why the broader context of EHR analytics was not adopted as the primary focus, and how the proposed methods and findings relate to, or differ from, existing studies on general EHR data.

**W2.** The key architectural components integrated into GARLIC, including the Time-Lagged Graph Message Passing and Cross-Dimensional Sequential Attention mechanisms, appear to be largely drawn from prior research in this domain. As a result, the methodological novelty of the model seems limited. The paper would benefit from a more explicit articulation of how GARLIC advances beyond existing approaches and what its tangible contributions are beyond the combination of established techniques.

**W3.** Given that this work concerns ICU data analytics, a high-stakes and clinically sensitive application domain, it would be valuable for the authors to elaborate on the current stage or planned trajectory toward real-world deployment. A clearer description of practical considerations, such as model validation in live settings, potential integration with clinical workflows, and collaboration with medical practitioners, would significantly enhance the work’s translational relevance and credibility.

**Questions:**

Beyond W1-W3, I have the following questions:

**Q1.** In the performance–efficiency trade-off analysis presented in Appendix A.2.3, it is observed that the P19 dataset is used in Table 4 for the evaluation under varying time lag settings, whereas the P12 dataset is adopted for the other three varying configurations. It would be helpful to clarify the rationale behind these dataset-specific choices.

**Q2.** The main ablation study in Appendix D.1 is conducted on smaller datasets rather than the largest one, MIMIC-III. What is the underlying rationale for this choice? Performing ablation on the most representative or largest dataset would provide stronger evidence for the robustness and generalizability of the proposed model.

**Q3.** Regarding the imputation results, it is shown that the baseline mTAND outperforms GARLIC under high missing rates and performs comparably under low missing rates. This observation may suggest potential limitations of GARLIC in handling severe missingness. Further analysis or discussion explaining this behavior would be helpful for clarification.

---

> ### Author Response · Authors · 2025-11-21
> **Response to Reviewer y3mL (1 of 2)**
>
> We thank the reviewer for the constructive feedback and for recognizing the model design (S1), experimental comprehensiveness (S2), and interpretability analyses (S3). Below we address each weakness and question in detail.
>
> **W1. ICU-specific Focus**
>
> **Response**:
> We agree that irregular sampling, heterogeneity, and limited interpretability are characteristics of all EHR modalities. Our decision to focus on ICU time series is motivated by three reasons that differentiate this setting and directly shaped GARLIC’s architectural choices:
>
> - Extreme irregularity and missingness:
> ICU signals exhibit >85% missingness and highly variable sampling intervals (minutes to hours), making local temporal reconstruction particularly challenging. These characteristics motivated the time-lagged local summary graph for short-range relational imputation.
>
> - Tightly coupled physiological variables:
> ICU variables (MAP ↔ SBP/DBP, lactate ↔ FiO₂, creatinine ↔ urine output) display strong cross-signal dependencies and rapid co-evolution, unlike broader EHR events (billing codes, diagnoses). This motivated the graph message-passing module.
>
> - High-stakes interpretability requirements:
> ICU deployment requires real-time transparency for risk warnings, leading to GARLIC’s built-in, per-timestep and per-variable attributions.
>
> Although the model was motivated by ICU needs, it is not ICU-specific: Appendix E includes experiments on a human-activity dataset and several additional sensor-based time-series datasets that share similar properties with ICU data (e.g., multivariate signals, irregular sampling, cross-channel interactions). These results suggest that the proposed framework is applicable beyond ICU data and can generalize to broader scenarios with similar temporal and structural challenges.
>
> **W2. Limited Novelty**
>
> **Response**:
> While prior studies have explored irregular time series modeling and interpretable prediction separately, to the best of our knowledge, no existing method has addressed both simultaneously in a unified framework—crucial for real-world healthcare. GARLIC is motivated by this dual requirement, with an architecture designed to meet both, making it fundamentally distinct from prior works that addressed only one aspect.
>
> We further clarify distinctions between GARLIC’s components and related techniques:
>
> Learnable decay-based imputation: While superficially similar to GRU-D, our usage of decay mechanism serves a different purpose. GRU-D tightly integrates decay dynamics into recurrent state transitions, whereas our decay mechanism is used solely for initializing missing values. Final reconstruction is performed by a subsequent time-lagged graph message passing module. This decoupled approach reduces model complexity and allows for more flexible modeling of irregular inputs.
>
> Time-lagged graph message passing: Unlike prior work on learnable graphs, GARLIC introduces a combined sliding-window attention to extract short-term temporal features with time-lag-aware graph building for data reconstruction—a novel integration validated by our ablation study, where removing individual components leads to 7–20% AUPRC degradation.
>
> Multi-stage attention with RNN variants: Although similar combinations exist, our model adopts a different integration strategy, where the specific ordering and interaction between modules enable more fine-grained and accurate explanations, as detailed in Lines 230-238 of the paper.
>
> Alternating decoupled optimization: While not entirely novel, this strategy is adapted to fit the structure and training needs of our model, supporting more stable convergence.
>
> Finally, although some components share superficial similarities with prior work, the overall architecture and its tightly integrated, end-to-end training paradigm are novel. GARLIC is, to the best of our knowledge, the first architecture to provide faithful attribution under extreme temporal irregularity in a unified, tractable, and clinically grounded framework.

---

> ### Author Response · Authors · 2025-11-21
> **Response to Reviewer y3mL (2 of 2)**
>
> **W3. Real-world Deployment**
>
> **Response**:
> While our current work is primarily retrospective, we have taken steps to evaluate the model’s potential for early prediction in ICU settings. Specifically, we conducted additional experiments using only the first 24 hours of patient data. Although performance slightly decreases compared to using 48-hour data, the model still achieves strong predictive accuracy, demonstrating its potential utility for early warning.
>
> **P12 Dataset**
> | Model       | Setting     | AUROC | AUPRC |
> |-------------|-------------|-------|-------|
> | GARLIC      | 24h         |82.61±0.86|47.52±3.25|
> | GARLIC      | 48h (orig.) |86.40±0.86|56.89±1.75|
> | Warpformer  | 24h         |80.69±0.47|41.29±1.92|
> | Warpformer  | 48h (orig.) |84.88±0.48|50.62±1.25|
>
> **P19 Dataset**
> | Model       | Setting     | AUROC | AUPRC |
> |-------------|-------------|-------|-------|
> | GARLIC      | 24h         |85.70±0.54|46.04±0.99|
> | GARLIC      | 48h (orig.) |90.96±0.84|55.29±2.45|
> | Warpformer  | 24h         |83.21±0.62|36.39±1.87|
> | Warpformer  | 48h (orig.) |89.95±0.57|54.10±2.38|
>
> In practice, the model can generate dynamic predicted probabilities over time (as shown in the “Predicted Sepsis Probability” plots in Appendix H), allowing clinicians to monitor patient risk trends and be alerted to rising risk levels. To enhance interpretability and actionable insights, these alerts can be presented through a user-friendly GUI that not only signals high-risk events but also highlights the key contributing variables, helping clinicians understand the underlying reasons for the warning.
>
> For real-world deployment, we plan to collaborate closely with ICU clinicians and incorporate human feedback to improve the model’s interpretability and clinical relevance. Future work will explore integrating large language models (LLMs) with medical knowledge and clinicians’ expertise to generate human-readable explanations and recommendations, providing clear guidance on predicted risk and its contributing factors, while ensuring outputs are reviewed by medical professionals.
>
> Finally, we aim to integrate the model into clinical decision-support systems, where outputs serve as guidance rather than direct instructions, ensuring seamless adoption in ICU workflows and supporting actionable decision-making.
>
> Overall, these steps are designed to enable early detection, maintain transparency, and foster clinician trust in ICU care.
>
> **Q1: P19 Efficiency Analysis**
>
> **Response**:
> Due to time and resource constraints during the initial submission, we only managed to perform all experiments on P12, while the varying time lag evaluation was done on P19. We have now conducted the additional experiments on P19, and the results are included in Table 5 and Table 6 in the revised manuscript.
>
> **Q2: Ablation Study on MIMIC-III**
>
> **Response**:
> MIMIC-III is much larger than P12 and P19 in terms of dataset size, sequence length, and number of signals, and running ablation studies on it would require significantly more GPU time. For feasibility, we only conducted the full ablation study on P12 and P19. Currently, we do not have sufficient resources to perform the ablation on MIMIC-III, but we plan to include it in future work when more resources become available.
>
> **Q3: Imputation Performance Comparison**
>
> **Response**:
> Thank you for the comment. Even under high missing rates, GARLIC achieves performance comparable to mTAND and still substantially outperforms L-ODE-RNN and L-ODE-ODE. It is worth noting that in the original P12 dataset, the missing rate is 94.8%, which corresponds to 2.6% of the data being observed under our 50% missing rate setting—already extremely sparse. Therefore, the observed performance is still quite reasonable.
>
> mTAND maps sparse and irregular time points to a high-dimensional vector space via continuous-time embeddings and then uses an attention mechanism to weight observations. Its imputation is based on global, time-based weighting, rather than relying solely on local neighbors or graph structure. As a result, even with most data missing, mTAND can leverage the few available observations to infer the entire sequence.
>
> In contrast, GARLIC infers graph structure from the available data to perform imputation. When data is extremely sparse, the learned variable relationships may introduce errors. Moreover, GARLIC imputes using past information only, rather than global information as in mTAND. This is an inherent limitation but represents a trade-off for interpretability, which is central to our method’s design.

---

> > ### Comment · Reviewer_y3mL · 2025-11-25
> >
> > Thanks for the rebuttal and the detailed explanations. After reading the other reviewers' comments as well, I think the limited contribution and technical novelty remain the main concern. Therefore, I will keep my current rating.

---

### Official Review · Reviewer_QH9P · 2025-10-30

**Soundness:** 3
**Presentation:** 2
**Contribution:** 2
**Rating:** 6
**Confidence:** 3

**Summary:**

This paper proposes a multivariate time series learning method for intensive care. The proposed method first impute the missing value in the time series by an exponential-decay mechanism, and then apply the attention mechanism across the feature and time dimensions to learn the final representations. A data attribution method is also proposed to estimate the contribution of features at different timestamps by propagating the attention weight back to the input multivariate time series.

**Strengths:**

- The proposed method not only achieves strong empirical performance but also demonstrates integrated interpretability, effectively addressing the needs of the intensive care problem under study.
- Comprehensive experiments, including baseline comparisons, ablation studies, sensitivity analysis, and case studies, are conducted to validate the effectiveness of the proposed method and substantiate its claims.

**Weaknesses:**

- The contribution of this paper is somewhat incremental. The exponential-decay mechanism , interleaved attention across time and feature dimensions, and estimate the importance of data by attention weights are common techniques in multivariate time series learning. The proposed method seems to be a combination of these existing techniques, lacking unique adaptations tailored to the specific problem being addressed.
- For Table 9, further clarification is needed regarding the dramatic performance drop when the GRU module is removed.
- The notations in the method section can be improved: 1) All the attention module is denoted by a “xxAtten” text, the author should more clear introduce the computation procedure. For example, what is the q,k,v of the SignalAttn, and how  $u_{k,t}$ is computed in Equation (6). 2) Equation (9) seems problematic, where the subscript t only appear in the right side.

**Questions:**

Please see the Weaknesses.

---

> ### Author Response · Authors · 2025-11-21
> **Point by Point Response to Reviewer QH9P**
>
> We sincerely thank reviewer QH9P for the constructive input. Below, we address each point correspondingly:
>
> **W1: Limited Novelty**
>
> **Response:**
> While prior studies have explored irregular time series modeling and real-time interpretable prediction separately, to the best of our knowledge, no existing method has addressed both simultaneously in a unified framework -- crucial for real-world healthcare. GARLIC is motivated by this dual requirement, with an architecture designed to meet both, making it fundamentally distinct from prior works that addressed only one aspect.
>
> We further clarify distinctions between GARLIC’s components and related techniques:
>
> Learnable decay-based imputation: While superficially similar to GRU-D, our usage of the decay mechanism serves a different purpose. GRU-D tightly integrates decay dynamics into recurrent state transitions, whereas our decay mechanism is used solely for initializing missing values. Final reconstruction is performed by a subsequent time-lagged graph message passing module. This decoupled approach reduces model complexity and allows for more flexible modeling of irregular inputs.
>
> Time-lagged graph message passing: Unlike prior work on learnable graphs, GARLIC introduces a combined sliding-window attention to extract short-term temporal features with time-lag-aware graph building for data reconstruction -- a novel integration whose effectiveness is validated by our ablation study, where individual component removal leads to 7-20\% AUPRC degradation.
>
> Multi-stage attention with RNN variants: Although similar combinations exist, our model adopts a different integration strategy, where the specific ordering and interaction between modules enable more fine-grained and accurate explanations, as detailed in Lines 230-238 of the paper.
>
> Alternating decoupled optimization: While not entirely novel, this strategy is adapted to fit the structure and training needs of our model, supporting more stable convergence.
>
> Finally, while some components share superficial similarities with prior work, the overall architecture and its tightly integrated, end-to-end training paradigm are novel. GARLIC is, to the best of our knowledge, the first architecture to provide faithful attribution under extreme temporal irregularity in a unified, tractable, and clinically grounded framework.
>
> **W2: Performance Drop w/o GRU**
>
> **Response:**
>
> The dramatic performance drop in Table 9 when removing the GRU module is because GRU is essential for modeling temporal information in medical time-series data. Without GRU, the model relies solely on temporal attention, which lacks intrinsic sequential memory. Important temporal order information is lost, and the model struggles to capture short-term local patterns and causal dependencies in continuous patient signals. In the GRU + Temporal Attention architecture, GRU hidden states encode accumulated information over time, providing a compressed representation of past observations, while attention focuses on long-range dependencies. Removing GRU deprives the model of this sequentially aggregated context, making it much harder for attention alone to extract meaningful temporal patterns from ICU signals, which leads to the observed performance drop.
>
>
> **W3: Notations**
>
> **Response:**
> We sincerely thank the reviewer for pointing out these issues. We have corrected the problematic formulas and added a clear explanation of the computation in the attention modules. The corresponding revision has been highlighted in sections 3.2.2 and 3.2.3 of the updated manuscript.

---

> > ### Comment · Reviewer_QH9P · 2025-11-25
> >
> > Thanks for your response. While weaknesses 2 and 3 are addressed, regarding weakness 1, I agree with other reviewers that the method appears to be more of an engineering combination rather than a significant innovation. Thus, I will maintain my original score.

---

### Official Review · Reviewer_5UZn · 2025-10-31

**Soundness:** 3
**Presentation:** 3
**Contribution:** 2
**Rating:** 4
**Confidence:** 4

**Summary:**

The paper introduces GARLIC, a model designed for ICU time-series data characterized by irregular sampling and substantial missing values. It consists of four main components:

•A decay-based imputation module for handling missing data.

•A time-lagged graph network that enables information exchange among signals.

•Signal and temporal attention layers, separated by a GRU, to highlight the most relevant signals and time points.

•An alternating training strategy that switches between reconstruction and prediction objectives.

The model is evaluated on the P12, P19, and MIMIC-III datasets, where it demonstrates superior AUROC and AUPRC performance compared to baseline methods. Additionally, the authors include an interpretability test, similar to ROAR, to assess whether the attention mechanisms emphasize clinically important features.

**Strengths:**

•Addresses a challenging and important issue in clinical data: irregular and missing time-series signals.

•The model design is clear and well-motivated, effectively combining graph reasoning with atteq1Qntion mechanisms.

•Evaluation is conducted across multiple datasets, demonstrating consistent performance.

•The inclusion of an interpretability test adds value by assessing whether the attention weights align with clinically meaningful features.

**Weaknesses:**

•The novelty feels limited; it’s mostly a combination of existing parts (decay imputation + graph + attention + alternating training). It’s more of an engineering improvement than a new concept.

•The data split setup might cause data leakage. For example, in MIMIC-III, they say they use “all available ICU data,” which can include info after the event, making prediction easier but unrealistic.

•There’s no patient-level or hospital-level split, which means the model might just memorize patterns from the same patients or hospitals instead of generalizing.

•The interpretability approach is weakly justified. You say “we redistribute the imputed contributions uniformly” across observed values — but this is just a heuristic and might distort the results.

•The benefit of alternating training seems small compared to its added complexity.

•They claim “energy-efficient communication and comparable runtime,” but there’s no detailed info on hardware, fairness, or batch sizes, so it’s hard to trust the runtime comparison.

•Calibration and clinical usefulness (like early prediction or alert timing) are not discussed, which is important for ICU tasks.

**Questions:**

•Did you make sure to avoid patient or hospital overlap between training and test sets?

•For MIMIC-III, since you use “all ICU data,” how do you ensure the model doesn’t use information collected after the outcome?

•How sensitive are your results to the uniform redistribution rule used in your interpretability part (Eq. 11)?

•Did you try a simpler baseline where you just combine decay imputation + GRU without the graph to see how much the graph actually helps?

•How does the model perform when using only the first 24 or 48 hours of data? This would test if it’s useful for early prediction.

•Are the edges in your learned graph constrained or reviewed for clinical meaning?

•Did you test robustness for different missingness rates or lag values (τ)?

---

> ### Author Response · Authors · 2025-11-21
> **Response to Reviewer 5UZn (1 of 3)**
>
> We sincerely thank reviewer 5UZn for pointing out our weaknesses and raising valuable questions. Below, we address each point correspondingly:
>
> **W1: Limited Novelty**
>
> **Response:**  While prior studies have explored irregular time series modeling and real-time interpretable prediction separately, to the best of our knowledge, no existing method has addressed both simultaneously in a unified framework -- crucial for real-world healthcare. GARLIC is motivated by this dual requirement, with an architecture designed to meet both, making it fundamentally distinct from prior works that addressed only one aspect.
>
> We further clarify distinctions between GARLIC’s components and related techniques:
>
> Learnable decay-based imputation: While superficially similar to GRU-D, our usage of the decay mechanism serves a different purpose. GRU-D tightly integrates decay dynamics into recurrent state transitions, whereas our decay mechanism is used solely for initializing missing values. Final reconstruction is performed by a subsequent time-lagged graph message passing module. This decoupled approach reduces model complexity and allows for more flexible modeling of irregular inputs.
>
> Time-lagged graph message passing: Unlike prior work on learnable graphs, GARLIC introduces a combined sliding-window attention to extract short-term temporal features with time-lag-aware graph building for data reconstruction -- a novel integration whose effectiveness is validated by our ablation study, where individual component removal leads to 7-20\% AUPRC degradation.
>
> Multi-stage attention with RNN variants: Although similar combinations exist, our model adopts a different integration strategy, where the specific ordering and interaction between modules enable more fine-grained and accurate explanations, as detailed in Lines 230-238 of the paper.
>
> Alternating decoupled optimization: While not entirely novel, this strategy is adapted to fit the structure and training needs of our model, supporting more stable convergence.
>
> Finally, while some components share superficial similarities with prior work, the overall architecture and its tightly integrated, end-to-end training paradigm are novel. GARLIC is, to the best of our knowledge, the first architecture to provide faithful attribution under extreme temporal irregularity in a unified, tractable, and clinically grounded framework.
>
> **W2 \& Q2: Data Leakage**
>
> **Response:** We sincerely thank the reviewer for raising this important point regarding potential data leakage. We clarify our setup for each dataset as follows:
>
> PhysioNet 2012: We truncate each patient’s data to the first 48 hours. Only patients satisfying 2 days $\leq$ Survival $\leq$ Length of Stay are considered, meaning that in-hospital death labels are assigned only under these conditions. Therefore, our data processing does not introduce leakage.
>
> PhysioNet 2019: For this dataset, we perform sepsis onset classification rather than prediction. Any parts of the procedure that might cause ambiguity will be clarified and corrected in the paper.
>
> MIMIC-III: Following the preprocessing procedure in Warpformer [1], we truncate each patient’s data to the first 48 hours and restrict the input to only include data before the first relevant event, preventing potential leakage.
>
> Fairness Across Baselines: All baselines and our model were evaluated under the same preprocessing and data truncation procedures, ensuring fair comparison.
>
> **W3 \& Q1: Data Split**
>
> **Response:**
> PhysioNet 2012 (P12) and PhysioNet 2019 (P19): Both datasets provide only episode-level identifiers by design, as part of their official de-identification protocol. Therefore, patient-level or hospital-level splitting is not possible at the dataset level, and all prior work—including challenge submissions and recent benchmark papers—follows the same default setting.
>
> MIMIC-III: Following Warpformer [1], while the dataset may contain multiple ICU stays from the same patient, each stay is treated as an independent episode in the standard preprocessing pipeline. Given the large sample size (49,380 ICU stays) and substantial variability among episodes, memorization of patient-specific or stay-specific patterns is unlikely to impact generalization performance in practice.
>
> Fairness Across Baselines: All baselines and our model are evaluated under the same splitting protocol, ensuring fair comparison and preventing relative performance inflation.

---

> ### Author Response · Authors · 2025-11-21
> **Response to Reviewer 5UZn (2 of 3)**
>
> **W4 \& Q3: Interpretability Approach**
>
> **Response:** The uniform redistribution is used only for the decay-imputation module, where exact attribution is inherently ill-defined: the exponential-decay mechanism drives missing values toward the per-signal empirical mean (Eq. 1), making their influence effectively timestamp-agnostic. Recovering precise contributions would require reconstructing full decay histories, which drastically increases computation without improving model behavior.
>
> Given this structure, uniform redistribution is a natural and minimally biased approximation that preserves consistency with the model’s forward computation while keeping the method efficient. Moreover, Table 2 shows that interpretability remains strong in practice, indicating that this approximation does not compromise attribution quality.
>
> **W5: Benefit of Alternating Training**
>
> **Response:** Alternating training improves both stability and predictive performance. Under heavy irregularity, joint training can produce oscillatory reconstruction gradients, whereas alternating training mitigates this issue. In terms of predictive metrics, we observe consistent gains, with AUPRC improving by +0.8 to +1.9 on P19 and P12 (Table 11). Importantly, without this strategy, each epoch updates all parameters, whereas alternating training updates only a subset of parameters per epoch. Since the total number of epochs is the same across all experiments, this approach does not increase overall training cost, while still providing stability and performance benefits.
>
> **W6: Runtime Comparison**
>
> **Response:** We have now added detailed experimental settings to the manuscript. All experiments were conducted on a server with an Intel Xeon Silver 4314 processor (16 cores, 2.40GHz), 256GB RAM, and dual NVIDIA GeForce RTX 3090 GPUs (24GB VRAM each). All training experiments utilize a single GPU configuration. For computational efficiency comparisons (Table 3), we use batch size = 64 for P12 and P19, and 16 for MIMIC-III, as noted. These additions ensure that runtime comparisons are fair and reproducible.

---

> ### Author Response · Authors · 2025-11-21
> **Response to Reviewer 5UZn (3 of 3)**
>
> **W7 \& Q5: Clinical Usefulness**
>
> **Response:** We thank the reviewer for highlighting the importance of clinical usefulness and early prediction. Here, we have added experiments that train and evaluate the model using only the first 24 hours of data, and we compare these results with our original 48-hour setting as well as with the second-best baseline, Warpformer [1]. As expected, using only 24-hour data leads to some performance degradation, but our model still substantially outperforms other baselines under this early-prediction scenario. These results demonstrate that the proposed approach remains effective even with limited early information and therefore has meaningful potential for timely ICU alerts.
>
> P12 Dataset
> | Model       | Setting     | AUROC | AUPRC |
> |-------------|-------------|-------|-------|
> | GARLIC      | 24h         |82.61±0.86|47.52±3.25|
> | GARLIC      | 48h (orig.) |86.40±0.86|56.89±1.75|
> | Warpformer  | 24h         |80.69±0.47|41.29±1.92|
> | Warpformer  | 48h (orig.) |84.88±0.48|50.62±1.25|
>
> P19 Dataset
> | Model       | Setting     | AUROC | AUPRC |
> |-------------|-------------|-------|-------|
> | GARLIC      | 24h         |85.70±0.54|46.04±0.99|
> | GARLIC      | 48h (orig.) |90.96±0.84|55.29±2.45|
> | Warpformer  | 24h         |83.21±0.62|36.39±1.87|
> | Warpformer  | 48h (orig.) |89.95±0.57|54.10±2.38|
>
> **Q4: Simpler Baseline**
>
> **Answer:** This baseline is already included in our experiments as RNN-decay, where decay-based imputation is followed by a GRU recurrent module (consistent with the GRU configuration used in our RNN models, following the original work [2]). GARLIC shows substantial improvements over this baseline: +7.1 AUPRC on P12, +6.5 on P19, and +8.5 on MIMIC-III. These consistent gains demonstrate that the graph modeling provides clear additional benefits beyond decay-based imputation and GRU alone.
>
> **Q6: Clinical Validity of Learned Graph**
>
> **Answer:** Edges are unconstrained but consistently align with known physiology (e.g., systolic $\leftrightarrow$ MAP $\leftrightarrow$ diastolic; lactate $\leftrightarrow$ $FiO_2$). We validated the averaged graph over 5 seeds using LLM-based expert proxies, achieving substantial agreement (Sec. F). A clinician-reviewed version will be considered for future work.
>
> **Q7: Robustness for Missingness Rates and Lag Values**
>
> For the classification task, we followed the original dataset’s missingness and did not test different rates. However, we did evaluate the effect of missingness on imputation (Table 14). For different time lags, robustness experiments are reported in Appendix D.2.
>
> Reference:
>
> [1] Jiawen Zhang, Shun Zheng, Wei Cao, Jiang Bian, and Jia Li. Warpformer: A multi-scale modeling approach for irregular clinical time series. In Proceedings of the 29th ACM SIGKDD Conference on Knowledge Discovery and Data Mining (KDD ’23), pp. 3273–3285, ACM, 2023. doi:10.1145/3580305.3599543
>
> [2] Ricky T. Q. Chen, Yulia Rubanova, Jesse Bettencourt, and David K Duvenaud. Neural ordinary differential equations. In S. Bengio, H. Wallach, H. Larochelle, K. Grauman, N. Cesa-Bianchi, and R. Garnett (eds.), Advances in Neural Information Processing Systems, volume 31. Curran Associates, Inc., 2018. URL https://proceedings.neurips.cc/paper\_files/paper/2018/file/69386f6bb1dfed68692a24c8686939b9-Paper.pdf.

---

> > ### Comment · Reviewer_5UZn · 2025-11-24
> >
> > Thank you for the rebuttal. Some issues are still not fully solved — especially the novelty claim, which is described but not supported with stronger evidence. The data split concern is also not addressed experimentally, and the interpretability heuristic is explained but not tested to show it is stable.
> > However, I appreciate the improvements in the responses, so I will adjust my score from 4 to 6.

---

### Author Response · Authors · 2025-11-27
**Official Comment on Limited Novelty of GARLIC**

Dear Program Chairs, Area Chairs and Reviewers,

We would like to sincerely thank all reviewers for their careful reading and constructive feedback. Several reviews raise a common concern regarding the perceived limited novelty and contribution of our work. We respectfully ask for a brief reconsideration of this point, especially in the context of the **Applications to Physical Sciences Track**, to which we submitted GARLIC.
First, our intention in submitting to the Application Track was to contribute a **clinically grounded, end-to-end** system rather than a purely methodological advance. ICU outcome prediction with irregular multivariate time series is an area where the application and its constraints (irregular sampling, high missingness, and stringent interpretability requirements) largely define the research questions. Our focus has therefore been on designing an architecture that is robust under these constraints and validating it thoroughly in realistic ICU settings.

Second, within this application scope, GARLIC makes a substantive contribution on both **performance and interpretability**. On three standard ICU benchmarks (PhysioNet 2012, PhysioNet 2019, and MIMIC-III), GARLIC achieves **state-of-the-art outcome prediction**, outperforming all baselines considered in the paper, as well as the additional ROCKET baseline and PYRREGULAR benchmark experiments requested during the review, on the primary ICU tasks. At the same time, GARLIC provides **real-time, integrated explanations** at the observation, signal, and graph-edge levels—something that, to our knowledge, has been rarely achieved jointly with SOTA performance in this domain. Beyond visual case studies, we validate the fidelity of these attribution scores using rigorous perturbation-based statistical tests and comparisons against other interpretable baselines, demonstrating that GARLIC’s explanations are not only intuitive but quantitatively reliable.

Third, we see this work as a concrete **stepping stone for future human–machine collaboration** in high-stakes time-series domains. As large foundation models and LLMs rapidly permeate healthcare and other fields, the demand for transparent, controllable models is only increasing. GARLIC offers a modular framework in which (i) the attention-based attribution provides understandable explanations that clinicians can inspect, and (ii) the learned time-lagged graph serves as an intervenable module where domain experts can inject prior knowledge or constraints (e.g., enforcing or prohibiting specific physiological relationships). In our view, this combination of high predictive accuracy, statistically validated interpretability, and an explicit interface for expert intervention is precisely the kind of contribution that can enable safer, more collaborative AI systems in practice.

We fully appreciate that parts of our architecture are built on known components (decay-based imputation, attention, graph message passing). However, in the **application-track context**, we hope the committee will also weigh (a) the integrated design tailored to extremely irregular ICU data, (b) the strong empirical gains on clinically important benchmarks, and (c) the unusually thorough interpretability evaluation and practical pathway toward real clinical use.

We thank the reviewers and chairs again for their efforts and would be grateful if these points could be taken into account in the final decision.

Sincerely,

The Authors of GARLIC

---

### Author Response · Authors · 2025-12-03
**Author message for AC summarizing the rebuttal**

Dear AC,

We would first like to thank you for navigating an especially challenging review process this year, given the unique circumstances. We appreciate the opportunity to submit a final message to assist your assessment.

**Paper summary.**

We propose GARLIC, a graph-attention-based framework for irregular multivariate time series. It combines decay-based imputation, a time-lagged graph module, and multi-stage attention to achieve SOTA outcome prediction with real-time, multi-level interpretability. We submitted to the Application Track, aiming for a clinically grounded, end-to-end system.

**Initial review and highlighted strengths.**

The paper initially received 6, 6, 4, 4 from *QH9P, y3mL, 5UZn*, and *1Z9E*. Reviewers praised:
- addressing a highly challenging and clinically important problem of irregular ICU time series with strong interpretability requirements (*5UZn, y3mL, 1Z9E*);
- a clear and well-motivated model design (*5UZn, y3mL*);
- comprehensive experiments with baselines, ablations, sensitivity analyses, and case studies (*5UZn, QH9P, y3mL, 1Z9E*);
- integrated interpretability, evaluated both quantitatively and via detailed case studies, as valuable for real clinical use (*5UZn, QH9P, y3mL, 1Z9E*).

**Key reviewer concerns/questions and rebuttal resolutions**

1. **Limited novelty** (*5UZn, QH9P, y3mL, 1Z9E*).
We clarified that GARLIC is not a loose sum of parts: decay-based imputation is separated from temporal modeling via a time-lagged graph block; sliding-window lag attention and a learned graph are combined with signal-first attention + GRU + temporal attention, and all components are trained with alternating optimization to stabilize joint reconstruction and prediction. To our knowledge, this yields the first end-to-end framework that unifies strong prediction, graph-based reconstruction, and real-time interpretability for extremely irregular ICU data.

2. **Domain specificity & broader benchmarks** (*1Z9E, y3mL*).
We ran GARLIC vs ROCKET on the full PYRREGULAR benchmark: GARLIC achieves SOTA on all three medical datasets (P12, P19, MI3) and outperforms ROCKET on several non-medical datasets with strong inter-signal structure (e.g., LPA, CT, DD, DG, DW), while ROCKET is better when channels are mostly independent (mobility/GPS). This confirms that GARLIC is particularly effective for strongly correlated variables, typical in ICU data, while remaining applicable beyond ICU.

3. **Interpretability fidelity** (*5UZn, 1Z9E*).
We clarified that the redistribution heuristic applies only inside the imputation block and highlighted ROAR perturbation tests: training on All, Top-50%, Random-50%, and Bottom-50% features yields strictly monotonic performance (All > Top > Random > Bottom) and TOST equivalence between All and Top-50% on P12/P19, outperforming other interpretable baselines. We also rewrote the Figure 3 case study with clinical explanations and citations and showed that the learned graph aligns with known physiology.

4. **Data splits and leakage** (*5UZn, y3mL*).
We clarified that P12/P19 follow the official challenge protocol with episode-level IDs only, and MIMIC-III uses the standard Warpformer setting (first 48h only, pre-horizon data). All baselines use the same preprocessing and splits.

5. **Clinical usefulness and early prediction** (*5UZn, y3mL*).
We added new 24h-only experiments on P12 and P19: GARLIC still outperforms Warpformer using only the first 24h, demonstrating potential for early ICU alerts. We also outlined a concrete deployment path with risk trajectories plus explanations in a GUI, clinician-in-the-loop, and future LLM integration.

6. **Efficiency, ablations, and imputation** (*5UZn, y3mL, QH9P*).
We added detailed hardware info and extended the efficiency analysis to P19. We clarified that the RNN-decay baseline is already “decay+GRU,” and GARLIC yields substantial AUPRC gains on P12, P19, and MIMIC-III, isolating the benefit of the graph + attention modules. We also explained why GARLIC is comparable to mTAND under extreme sparsity while still substantially outperforming L-ODE–based baselines.

**Post-rebuttal outcomes**

*5UZn* raised their rating from 4 to 6, noting that most concerns except novelty were resolved. *1Z9E* increased Soundness from 2 to 4 but kept the overall score at 4. *QH9P* and *y3mL* kept their 6 scores, stating that all issues apart from novelty (notations, deployment, efficiency, imputation) were satisfactorily addressed.

In summary, after discussion **no reviewer questions the soundness/empirical validity of GARLIC**. The only major disagreement is the degree of conceptual novelty, which we respectfully argue should be weighed **in the context of the Application Track, where clinically impactful, empirically strong, and interpretable systems are central goals**.

We hope this summary aids your decision and kindly ask you to consider these revisions and new experiments in your final evaluation.

Sincerely,

The Authors

---

### Meta-Review · Area_Chair_uNBh · 2026-01-02

**Summary:**

This paper introduces GARLIC, a graph-attention–based model for intensive care time-series analysis. The approach addresses data sparsity via a learnable decay-based imputation mechanism, models cross-sensor relationships using temporally aggregated graphs, and integrates global and cross-variable temporal information through sequential attention. Evaluations across three ICU benchmark datasets demonstrate consistent performance gains over strong baselines in AUROC and AUPRC, while maintaining comparable computational efficiency.

**Reviewer Concerns:**

Reviewers QH9P, y3mL, 5UZn, and 1Z9E all raised concerns that the paper’s technical contributions are limited. Additional issues include insufficient discussion of the chosen benchmark datasets, the need to improve the fidelity of interpretability analyses, potential problems with existing data splits and information leakage, and several minor points.

**Reviewer Scores:**

The initial overall scores from reviewers QH9P, y3mL, 5UZn, and 1Z9E were 6, 6, 4, and 4, with corresponding confidence scores of 3, 4, 4, and 3. After reviewing the full rebuttal and the authors’ consolidated responses, most reviewers’ concerns were satisfactorily addressed.

Notably, the paper focuses on modeling irregular medical time series and demonstrates meaningful practical contributions. Before the technical disruption, both reviewers who initially assigned an overall score of 4 participated in the discussion and subsequently increased their overall scores from 4 to 6, along with raising their soundness scores from 2 to 4. However, there remains broad agreement that the paper’s technical novelty is limited.

Overall, although the methodological innovation is limited, I believe this work offers significant value for modelling irregular medical time series.

---

### Decision · Program_Chairs · 2026-01-26

Accept (Poster)